# Directional flows using capillary assembly of photo-deformable colloidal particles at water-air interfaces

David Urban ®[1,2,5] ✉, Marcel Rey ®[3,4], Antonio Ciarlo ®[4], Marie Friederike Schulte ®[3], Emiliano Descrovi ®[2] ✉ & Giovanni Volpe ®[4]

Colloidal particles at liquid interfaces experience long-ranged capillary interactions, whose magnitude and directionality depend on the particle shapes. When particle shapes are determined by fabrication or synthesis, the resulting shape-mediated interactions are predefined and often lead to the formation of persistent interfacial structures. Here, we introduce polymer particles at water-air interfaces whose shape and, therefore, interactions can be altered by illumination with polarized light. Specifically, we selectively trigger capillary self-assembly by anisotropically deforming the particles at the interface. Intriguingly, further deformation of already assembled particles induces sustained interfacial flows with velocities of up to 90 μm/s. Benefitting from polarization-defined deformation directions, we create flow-patterns that do not simply follow the illumination intensity pattern, such as shear flows along a single rectangular illumination stripe. We anticipate that this interplay between photo-deformation and capillary interactions of particles will enable various forms of mixing, manipulation, and assembly of soft matter at liquid interfaces.

Capillary interactions are a fundamental phenomenon at liquid interfaces. In fact, when small objects are adsorbed to an interface, they deform it. These deformations increase the surface tension and are therefore energetically unfavorable. As a consequence, lateral forces between the objects emerge to minimize the surface tension and the surface deformations. This mechanism enables the emergence of intriguing natural phenomena. For example, some insects can climb the slippery curved slopes at a pond edge by adopting quasistatic body postures at the air–water interface[1]. In doing so, they actively create surface deformations that interact with the external edge meniscus so that they are propelled uphill. Biological entities can also use surface deformations to interact directly with each other, for example, by means of anisotropic shapes creating anisotropic interface deformations resulting in long-ranged, oriented capillary attraction forces.

Correspondingly, ordered capillary assemblies have been reported on several length scales, including oval beetle bodies[2], ellipsoidal mosquito eggs[3] and boomerang-shaped *Ruppia maritima* pollen[4].

Regarding artificial systems, shape-mediated capillary forces were first demonstrated on the mesoscale by guiding the ordered interfacial assembly of millimetric synthetic particles with anisotropic geometries and patterned hydrophilicity into well-defined static patterns[5]. Building upon this concept, tunable systems have recently been reported. Using temperature[6,7], magnetic fields[8], or light[9,10], the mutual interaction and assembly of mesoscale objects was controlled directly at the liquid interface. In all these works, thin, sheet-like rafts were employed, with lateral sizes ranging between 50 μm and several millimeters. The rafts relied on significant contour undulations stemming from well-defined wrinkles, folds, or bending

[1]Department of Electronic Systems, Norwegian University of Science and Technology, O.S. Bragstads plass 2b, Trondheim, Norway. [2]Dipartimento di Scienza Applicata e Tecnologia, Politecnico di Torino, Corso Duca degli Abruzzi 24, Torino, Italy. [3]Institute of Physical Chemistry, University of Münster, Corrensstr. 28/30, Münster, Germany. [4]Department of Physics, University of Gothenburg, SE-, Gothenburg, Sweden. [5]Present address: SINTEF Digital, P.O. Box 124 Blindern, Oslo, Norway. ✉e-mail: david.urban@sintef.no; emiliano.descrovi@polito.it

to pin the liquid interface to different heights and to induce large, tunable interface deformations.

Downscaling this shape-mediated capillary control to colloidal particles, with sizes between 100 nm and several μm, is of great interest considering the prospect of assembling and manipulating large numbers of particles simultaneously, but involves the challenge of controlling the interface deformation geometries at much smaller scales. Thus, researchers have studied the capillary interactions of interface-adsorbed particles with anisotropic shapes such as cylinders[11], curved disks[12], cubes[13], or ellipsoids[14], which can be prepared by methods involving, e.g., photolithography[11,12], hematite synthesis[13,15], or mechanical stretching[16,17]. In fact, on the scale of a colloidal particle, a smooth prolate ellipsoidal shape can be sufficient to induce a sizeable near-quadrupolar interface deformation around a particle due to the constant wetting angle[18,19]. Minimizing these quadrupolar deformations, particles were then reported to assemble, for example, in large-scale tip-to-tip or side-by-side stackings[14,19–22]. However, since these works relied on prefabricating particles with fixed shapes, the assembled particle phases were static. Versatile tunable assembly phase changes have been recently predicted in simulations using an external control mechanism based on magnetic fields to tilt magnetic ellipsoidal particles at the liquid interface, thereby tuning the induced interface distortions and particle interactions[23,24]. Yet, experimental realizations of tunable colloidal assemblies, based on direct control over shape-mediated capillary forces, have so far remained elusive.

Here, we experimentally introduce a 2D-confined colloidal system that permits us to directly control capillary inter-particle attraction forces and to generate sustained flow patterns. The system consists of colloidal azopolymer particles adsorbed to an air–water interface, whose shape can be altered using polarized visible light. This effect, which can be anisotropic and polarization-dependent[25], has frequently been used in nanofabrication contexts to optically inscribe surface topographies in thin films[26–28] or to reconfigure microscale features and particles bound to solid substrates[29–32], embedded in soft matrices[33–35], or within microporous films[36,37]. Here, we start by extending this phenomenology to the interface-adsorbed, mobile azopolymer particles, inducing both anisotropic and in-plane isotropic deformations depending on the illumination's polarization. We then show how the resulting shape-mediated capillary effects permit local assembly and disassembly of particles. Finally, the continuous deformation of local capillary assemblies in denser particle settings is discovered to enable steady particle flows, whose intriguing properties can be controlled by simple intensity patterns with defined polarization.

## Results

### Shape-morphing particles at an air–water interface

We start by considering the behavior of single azopolymer particles adsorbed to a liquid interface when subjected to illumination with polarized visible light. Specifically, we show that these particles can be deformed into anisotropic shapes directly at the air–water interface by laser illumination with suitable polarization, complementing the studies that have shown this effect on dry substrates[29,31,32].

The photo-deformable azopolymer particles were prepared by a solvent evaporation method, using the commercial light-responsive polymer poly[(Disperse Red 1 methacrylate)-co-(methyl methacrylate) pDR1m-co-mma (Fig. 1a). In brief, a chloroform solution of the polymer was added to an aqueous solution containing polyvinylpyrrolidone (PVP) and emulsified by sonication while the chloroform was continuously evaporated (see Methods for details). This generated a polydisperse set of azopolymer particles, sterically stabilized by PVP, with diameters up to 3 μm (Supplementary Fig. 1a). These particles were spread at the air–water interface in a sealed sample cell, where their movement was confined to two dimensions. PVP was used as a steric stabilizer to promote particle adsorption to the interface and to

prevent aggregation upon adsorption. Thus, the particles were predominantly found at the interface plane (as shown by a confocal microscopy z-stack, Supplementary Fig. 2), reflecting their successful adsorption, and exhibited lateral Brownian motion in agreement with previous reports[38,39]. Using gel-trapping, we measured a three-phase contact angle (CA) of 104° ± 6° (Supplementary Fig. 3). Optical deformation was induced by illuminating the particles with an expanded green laser (λ = 532 nm) while observing them under an inverted optical microscope with white background illumination (Supplementary Movie 1).

When using a linearly polarized illumination beam, we were able to anisotropically stretch the particle along the polarization direction into a rod-like shape (Fig. 1b). Similar stretching behaviors have already been shown for similar azopolymer particles but on solid substrates[29,31,32]. Some studies also reported this transformation to be close to the uniaxial, isochoric one (i.e., the polymer being incompressible and the deformed particle being symmetric around the axis of polarization and elongation)[32,34].

Next, we observed circular polarization to induce in-plane isotropic flattening of particles, leading to an axially symmetric oblate, disk-like shape (Fig. 1c), again matching with previous studies focusing on immobile microstructures[30,35,40]. In Fig. 1c, note that the particle becomes less visible due to its flattening, reducing the lateral differences in optical path length (i.e., distance times refractive index, integrated along the optical axis) as well as the attenuation through the flattened particle, ultimately lowering imaging contrast.

Finally, elliptical polarization can be used to produce an intermediate result, i.e., a flattened ellipsoid with lower in-plane aspect ratio (Fig. 1d). Also this intermediate has been shown previously on solid substrates[30,37]. We note that, even in the theoretical case, such an ellipsoid will have three independently adjusted principal axes (no axial symmetry).

In Supplementary Fig. 4, the above examples are statistically quantified based on 20 individual deformation sequences for each case. We confirm that linear polarization induces the highest aspect ratios on average, followed by elliptical polarization, while circular polarization results in an in-plane aspect ratio close to one for the whole sequence. Further, we observe that all polarizations induce considerable flattening of particles, with relative area expansions of up to a factor of five for circular polarization. In this context, note that the underlying mechanism of photo-deformation in this polymer is based on cyclic *trans-cis-trans* isomerization induced in the pseudostilbene-type dye Disperse Red 1 (DR1)[41], which occurs due to the strong overlap of its *cis* and *trans* spectra at the employed wavelength[42]. While photo-thermal effects and the increase of *cis* isomer populations are known to accompany this effect for any polarization, the directionality of the material's photo-deformation is linked to a polarization-sensitive average reorientation of the dyes upon prolonged isomer cycling[25]. Hence, the different directionality of the particles' photo-deformation is the major change occurring when only the illumination polarization is altered in the experiments below.

### Optically controllable capillary interactions and assembly

Next, we show that shape-mediated, capillary-driven self-assembly—previously achieved using anisotropic particles that were fabricated in advance and then placed at liquid interfaces[11–14,19–22,43]—can instead be induced in situ via light-driven deformation of initially spherical particles directly at the interface. To do so, we study sample regions where pristine spherical particles diffuse freely and in close proximity along the interface. When shining elliptically polarized light on these particles, they deform into ellipsoids, approach each other and bind in a stable side-by-side configuration (Fig. 2a, i–iii and Supplementary Movie 2).

This behavior can be attributed to the anisotropic, saddle-like interface deformations, which are known to be induced by the

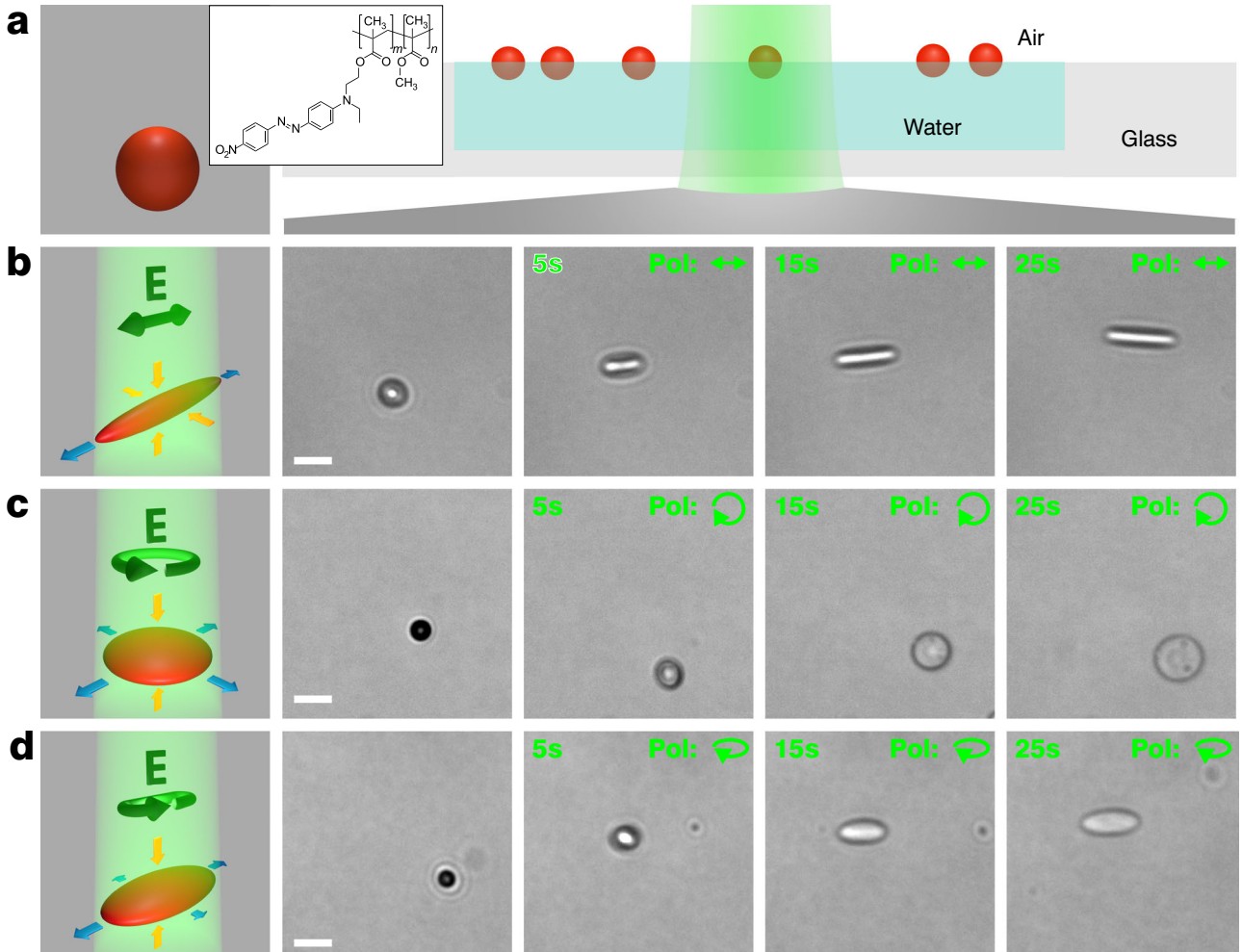

**Fig. 1 | Optical deformation of azopolymer particles at an air–water interface.**
**a** Scheme of a pristine spherical azopolymer particle (red), chemical structure of the polymer and scheme of the optical setup with particles adsorbed to the air–water interface. The laser beam (green) illuminates the sample from below.
**b**–**d** Schemes (left) and snapshots from Supplementary Movie 1 (right) showing the in situ deformation of particles with linearly (**b**), circularly (**c**), and elliptically (**d**) polarized light, leading to rod-like, disk-like, and flattened ellipsoidal particle shapes, respectively. Illumination intensity: $I$ = 219 W/cm², Scale bars (white): 5 µm. Blue/orange scheme arrows indicate respective expansion/contraction. Contrast levels in (**b**–**d**) are adjusted individually for each row.

constant three-phase contact angle around ellipsoidal particles[18,19], and are illustrated in the corresponding schemes shown in Fig. 2b, frames i–iii. The resulting long-range capillary forces, with quadrupolar symmetry, can significantly exceed the thermal energy $k_BT$[14,43,44], and lead to oriented particle assemblies. Thereby, the observed side-by-side configuration constitutes the minimum energy configuration for two ellipsoidal particles in contact[3,21,39]. The intermediate frames in Supplementary Fig. 5 show how the two particles rotate into this configuration upon contact[44,45]. The underlying capillary mechanism is overall surface energy minimization of the liquid interface, which can be achieved by superposing matching deformations, e.g., matching surface elevations or depressions[43].

Interestingly, the stable capillary bond between the particles can be reversed by a subsequent illumination with circular polarization (Fig. 2a, frames iv–v). This second illumination induces a biaxial flattening of the particles, which is applied on top of the previous deformation since azopolymer deformations are plastic, superposing processes[28,31,34]. Although the particle cross-sections do not become perfectly circular again during this process, the particles consistently disassemble during prolonged exposure.

This is likely due to a combined effect of in-plane ellipse aspect ratio reduction, intrinsically reducing the anisotropic interface deformation, and out-of-plane particle flattening (see schemes in Fig. 2b, frames iv–v). In fact, since the interface deformation causing the capillary attraction stems from contact line undulations around the physical particle, the maximal deformation height is approximately limited by the (gradually diminishing) particle height[43]. By flattening the particles, the increasingly sharp edge of the flattened particle associated to a diminishing out-of-plane curvature radius, is expected to become the dominant factor and to suppress the capillary attraction when the in-plane particle anisotropy is comparingly low.

Finally, this behavior is not restricted to pair-wise particle interactions. Additional insights can be obtained by considering regions containing multiple nearby particles. In such cases, capillary assembly and disassembly can involve larger particle ensembles (Fig. 2c and Supplementary Movie 3), driven by the long-range nature of the shape-mediated capillary forces, which have been reported to act over distances up to several tens of particle diameters[14,44,46]. While we observe pair-wise interaction up to 20 µm, we find that particles initially separated by up to 50 µm can become part of the same capillary assembly (Supplementary Fig. 6).

Additionally, the initial deformation may be performed using linearly polarized light, which deforms the particles into higher aspect ratio rod-like shapes (Supplementary Fig. 7 and Supplementary Movies

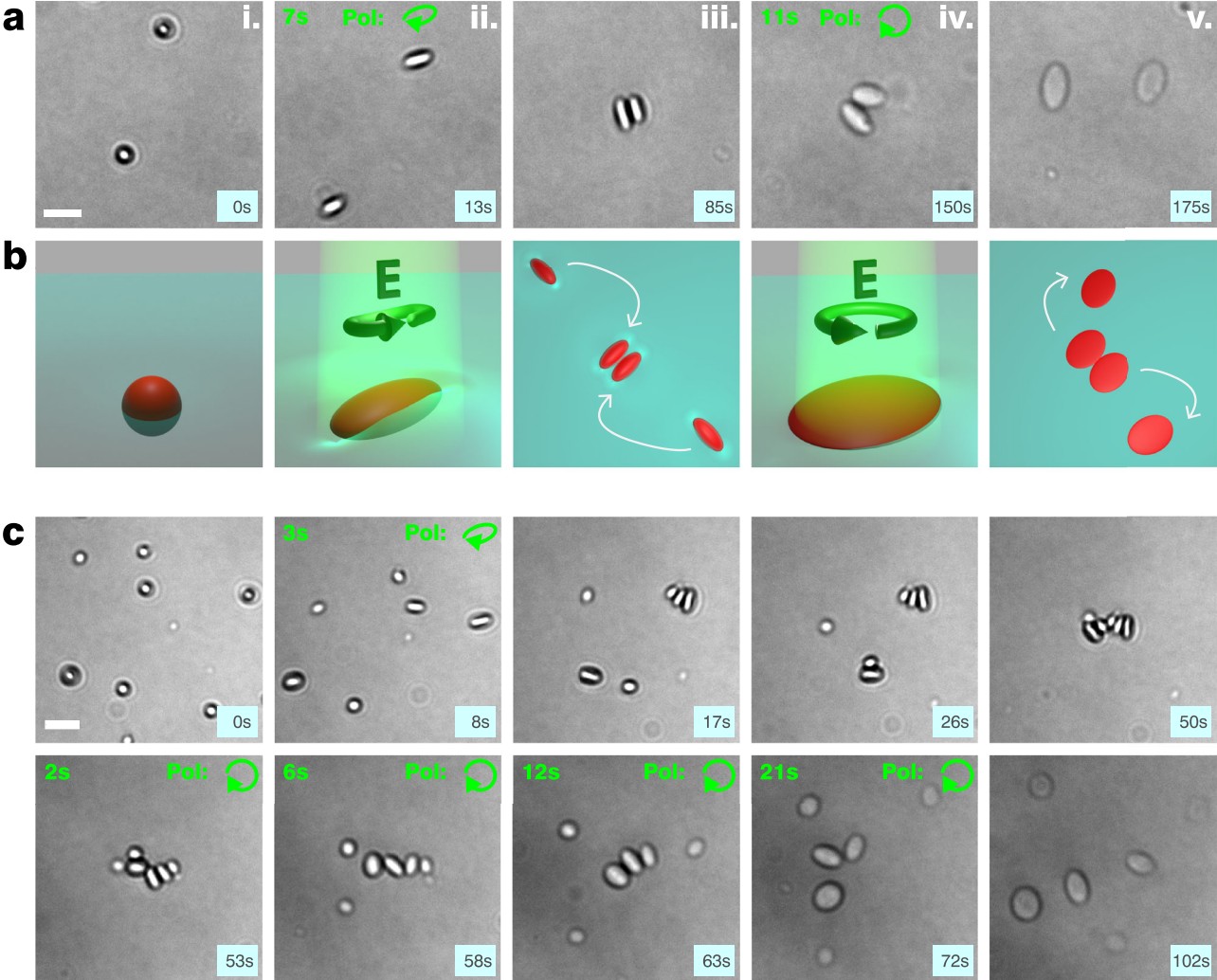

**Fig. 2 | Shape-mediated particle (dis)assembly induced by light. a** Experimental snapshots from Supplementary Movie 2, showing two undeformed, spherical azopolymer particles at the water–air interface (i), which are stretched into ellipsoidal particles upon irradiation with elliptically polarized light (ii), and subsequently bind to minimize the overall interface deformation (capillary effect) (iii). When they are deformed and expanded in the plane with circularly polarized light at a later stage (iv), they unbind as they become flattened particles (v). **b** Schemes depicting the corresponding stages with interface deformations emphasized for

illustration purposes. **c** Snapshots from Supplementary Movie 3, repeating the sequence in an area containing multiple particles. Note that particles continue to assemble due to their ellipsoidal shape, also after the elliptically polarized illumination has been switched off. They remain rigidly assembled until their shape is remodeled by illumination with circular polarization. Scale bars (white): 5 µm. Light blue squares in the lower right indicate total acquisition time, green font indicates respective time from illumination onset.

4, 5), although the disassembly time with circular polarization tends to increase in this case. Importantly, when circular polarization is used directly for the initial deformation, no assembly is induced, since the particles expand isotropically within the interface plane (Supplementary Fig. 8 and Supplementary Movie 6). Notably, in pre-assembled structures containing multiple particles, switching to circular polarization to reduce capillary attraction can trigger dynamic rearrangements. As the particles flatten and become increasingly rounded, the assemblies typically evolve into tip-to-tip configurations—the minimum energy configuration of multiple ellipsoids with lower-aspect ratios[39]—before eventually disassembling (Supplementary Fig. 9 and Supplementary Movie 7). In addition, the influence of particle size on the capillary forces becomes evident. Precisely, smaller particles are known to create smaller interface deformations and thus weaker attractive forces[43]. Therefore, they are not only pushed aside by larger particles during multi-particle assembly but also tend to dissociate more rapidly from other particles during disassembly (Supplementary Fig. 10 and Supplementary Movie 8).

## Continuously deforming capillary assemblies

We now report on the emergence of a collective, dynamic flow phenomenon arising from the combination of the previously described particle deformation and capillary assembly processes (Fig. 3 and Supplementary Movie 9). This intriguing effect was discovered when the particle density at the interface was significantly increased—from sparsely distributed particles in earlier sections (surface coverage: <5%) to a denser, near homogeneous surface coverage (Fig. 3a–c, frame i: surface coverage: 38%, see Methods), where the sterically-stabilized particles are still exhibiting Brownian motion. Upon illumination with a ~80 µm wide Gaussian laser beam profile polarized linearly along the $y$-axis, the particles in the entire illuminated area are stretched and start to self-assemble (Fig. 3a–c, frame ii). The assembled particles subsequently elongate further along the polarization direction and propagate their deformation along the growing capillary assembly (Fig. 3a, b, frame iii). At such high interfacial densities, both side-by-side and tip-to-tip configurations become energetically favorable, as each reduces interfacial deformation by superimposing

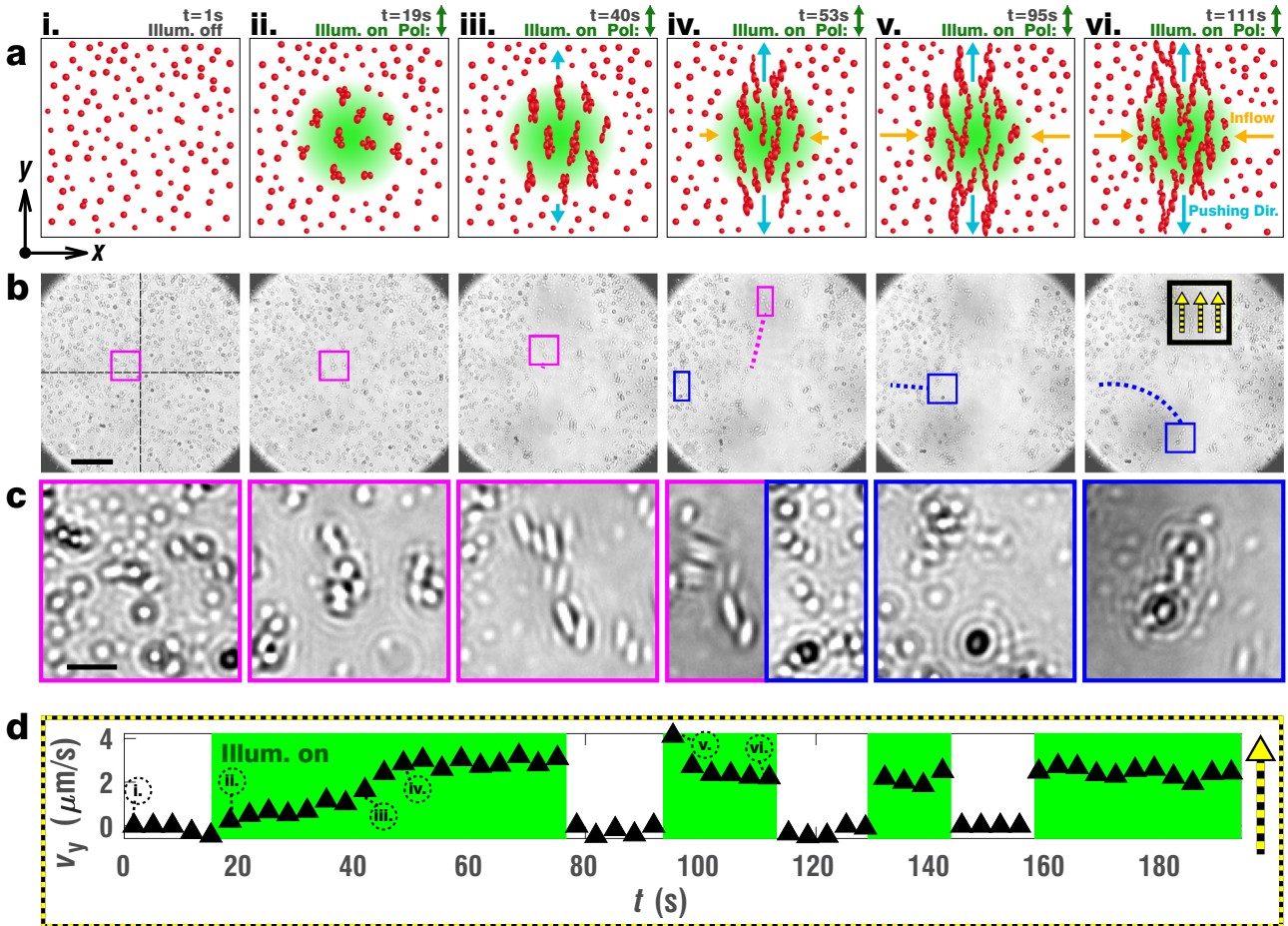

**Fig. 3 | Emergence of sustained symmetric particle flows under constant illumination. a** Sequential schemes showing the unperturbed interface crowded with freely diffusing pristine particles (i), which upon onset of y-axis linearly polarized illumination with a Gaussian illumination pattern start capillary assembly (ii), and under prolonged illumination keep deforming (iii), hence exerting a symmetric pushing force along the y-direction (light blue arrows) (iv). Expelled aggregates make space for fresh particles to get dragged into the illumination zone orthogonally (orange arrows) (v), fueling a sustained particle displacement flow (vi). Note that aggregates expelled from the illumination zone stay assembled since the particles are permanently deformed (Supplementary Fig. 11). **b** Snapshots of Supplementary Movie 9, corresponding to the schematically described steps. Scale bar (black): 30 μm **c** Magnified insets from (**b**). Magenta frames follow particles located slightly above the midline in the illumination zone before onset, which assemble, deform, and leave the field of view at the top due to the symmetric y-axis deformation. Blue frames, at a later stage, follow pristine peripheric particles which first get pulled in from the lower left below the midline, assemble, deform, and ultimately leave the field of view via the bottom in this case. Scale bar (black): 5 μm. **d** Graph showing the average y-axis velocity of particles passing the zone highlighted by a black/yellow frame in (**b**), as a function of illumination time. Encircled roman letters indicate the times corresponding to the six frames in (**a**–**c**). Green background indicates time intervals during which the sample is illuminated.

matching surface elevations and depressions of adjacent particles[21]. As the deformations propagate, assembled particles are seen to be pushed in y-direction towards the upper and lower boarders of the field of view and beyond (Fig. 3a, b, frame iv). The magenta inset, magnified in Fig. 3c, traces the course of large, well-recognizable pristine particles. On subsequent frames, these deforming particles self-assemble, deform further, and are pushed towards the upper border of the field of view by the collective symmetric deformation caused by all other assembled particles. Surprisingly, instead of a finite deformation, a continuous motion is generated (Fig. 3a, b, frames v-vi). In fact, the blue inset magnified in Fig. 3c displays fresh, peripheric particles at a later stage, which get pulled towards the zone of illumination along the x-direction to compensate for assembled particles leaving the zone along the y-axis. Entering the illumination zone, the pristine particles deform, assemble and contribute to sustaining the flow, initiating a stationary process.

To quantify this behavior, we measured the y-axis velocity of outflowing particle structures in the area contained by the black/yellow square in Fig. 3b, vi, using cross-correlation and maximum overlap fitting of subsequent frames (see Methods for details). The resulting velocities were plotted as a function of illumination time in Fig. 3d. After an initial phase where the particles are illuminated, but not yet assembled, the outflow velocity slowly rises to a constant value. When the illumination is turned off, the flow almost immediately stops. Subsequently, it resumes quickly at full speed when the illumination is switched on again. This may be explained with the particles already being assembled and thus being able to propagate their deformation instantly, in contrast to the dynamics at the first illumination onset.

At first glance, the transport phenomenon described above may appear similar to free particle flows along the interface. Indeed, several optically induced flow mechanisms capable of transporting particles or larger objects along liquid interfaces have been reported—the most relevant in this context being those based on *trans* to *cis* isomerization of azobenzene molecules or on photo-thermal phenomena[47,48]. For example, particles or oil droplets can exhibit directional locomotion along liquid interfaces when lateral gradients of *trans* and *cis* isomers with different hydrophilicity form on their surface[49,50]. Thermal gradients across particles and small objects can play a similar role[51,52]. Furthermore, Marangoni flows may be induced anywhere on a liquid interface using photo-thermal effects[53,54], or, e.g., by isomerization of

azobenzene-containing surfactant molecules[55–57], with the aim of displacing nearby particles.

To rule out such effects as the primary cause of the flow phenomena reported herein, several key observations can be considered. First, no locomotion is observed for sparsely distributed particles in the previous sections, although similar laser profiles were used. Therefore, flows arising from gradients across the particles themselves do not seem to drive the phenomenon. Secondly, the flow in our system stops immediately after the illumination is switched off. Yet, the relaxation of the *cis* population, accompanying the deformation, is known to take several seconds in the dark for the DR1-doped azopolymer[42,58], a timescale over which isomerization-related flows should proceed. Thirdly, upon close inspection of Supplementary Movie 9, one can easily observe the absence of individual diffusion for deformed particles when illumination is switched off, reflecting their assembled configuration. Indeed, the assembly could even be seen to extend far beyond the field of view in the *y*-direction (outflow direction) after the movie was acquired (Supplementary Fig. 11). Conventional flow mechanisms would then have to break up the rigidly assembled state in the illumination zone each time illumination is resumed, to cause smooth, free particle flows. Finally, and most importantly, the *y*-oriented directionality of the flow, which follows the particles' photo-deformation direction, contrasts with the rotationally symmetric illumination intensity profile (Gaussian beam). The full dependency of the local flow direction on the polarization is shown in the next section, amongst others, by freely rotating the outflow direction without modifying the illumination intensity pattern and by showing the absence of flow for circularly polarized illumination. Since photothermal and isomerization-based effects have no intrinsic directionality, this suggests a decisive role of the particles' polarization-directed photo-deformation in driving the flows.

On the other hand, the phenomenon cannot be explained by directional photo-deformation alone either, which is apparent when comparing the presented velocity graph to the well-studied time-dependent deformation effect in fixed azopolymer structures[31–33]. In these works, photo-deformation is shown to start deforming with a constant rate almost immediately when illuminated. Here, the macroscopically transmitted flow displacement initially arises only after particle assembly, tens of seconds later, when the particles can propagate their individual deformations. On the other hand, the referenced azopolymer deformations consistently saturate during prolonged exposure, upon reaching a maximal deformation. A film made of a single, large azopolymer structure would therefore be expected to cease flowing once the deformation in the illumination zone has saturated. In contrast, we observe that fresh particles can freely and constantly flow in from directions orthogonal to the outflow, reach the center of the illumination zone and efficiently evacuate heavily deformed particles as the capillary assemblies are remodeled. Thus, we conclude that it is the synergy of local, illumination-triggered particle assembly and polarization-dependent particle photo-deformation itself, which drives the directional flow.

### Polarization-controlled directionality and shear flows

Finally, we demonstrate how the sustained interfacial flows introduced above can be readily tuned by adjusting the illumination pattern and especially the polarization direction (Fig. 4a). This permits us not only to drive both symmetrical flows with arbitrary direction and pure shear flows, but also to thoroughly confirm the link between particle photo-deformation direction and flow behavior. For these experiments, we use azopolymer particles with a smaller average size (≤1.5 μm; Supplementary Fig. 1b, CA = 111° ± 4°, Supplementary Fig. 3d), which were observed to result in faster and more homogeneous flows. Instead of solely monitoring outflow velocity, we divide the field of view into regular tiles (Fig. 4b). This approach allows us to compute local velocities and extract entire flow fields through cross-correlation (see

Methods). An entire flowfield upon Gaussian illumination with *y*-oriented linear polarization is shown in Fig. 4c. If the polarization is *x*-oriented instead, an orthogonally oriented flow pattern is observed (Fig. 4d), which is a direct consequence of the polarization defining the anisotropic particle deformation direction. The original flows are shown in Supplementary Movie 10. Intriguingly, one can also alter the symmetric outflow axis dynamically during illumination, for example, by polarization rotation (Supplementary Movie 11 and Supplementary Fig. 12). Thus, not only does the flow directionality break with the rotational symmetry of the intensity pattern, but the directionality can also be changed without modifying the latter. Importantly, the same intensity pattern can even be shown without inducing any flow, when circularly polarized light is used (Supplementary Movie 12 and Supplementary Fig. 13). As shown in the previous sections, circularly polarized illumination causes in-plane isotropic, biaxial particle flattening, but no capillary assembly, which would allow the particles to propagate these deformations. We finally remark that biaxial expansion is also not compatible with the inflow of fresh, non-assembled particles along any in-plane axis on the interface, thus inhibiting persistent flows for circularly polarized illumination even in cases where the particles have previously been assembled using linear polarization (Supplementary Fig. 14). These observations corroborate that the flow profiles cannot be attributed to thermal or isomerization-driven effects, which occur also when using circular polarization, as discussed earlier.

It is also important to note that smooth and steady flows for linear polarization may not occur at any interfacial particle loading. Although the manual particle dispersion (see Methods) inhibits very precise adjustments, we highlight this by varying the parameter in coarse steps (via pre-dilution/concentration), leading to three extreme situations compared under *y*-oriented linearly polarized illumination (Supplementary Fig. 15 and Supplementary Movie 13, details in Supplementary Note 1). Crucially, in a fully packed particle monolayer where Brownian motion is severely hindered, the film deformation rate peaks rapidly but decays as the deformation saturates, and no flow is sustained (Supplementary Movie 13, right). This mirrors the immediate onset and saturation behavior of regular azopolymer deformation, as discussed previously. The transient velocity field for this deformation is also highly divergent, due to the lack of particle inflow. Similarly, a net positive divergence (i.e., higher total outflow than inflow) is observed when the starting situation are dense but still diffusing particles as in Fig. 4c, d. This is confirmed by explicit divergence computation (inset, Fig. 4d) and may be attributed to the particles' increasing surface area as they are stretched. Importantly, using a suitable dilution of particles with respect to this case, one can obtain situations where a steady particle flow is still occurring (Supplementary Movie 13, left), but where the contribution of divergence to the flow fields is rendered negligible (Supplementary Note 1). Such situations can be exploited to, for example, create velocity fields where the particles flow into the illumination zone faster than they exit, using a suitable illumination pattern (Supplementary Fig. 16 and Supplementary Movie 14). They can also give rise to shear patterns, as detailed below.

Shear flows were created using stripe-shaped illumination, where the linear polarization is oriented at 45 degrees relative to the stripe (Fig. 4e, f). In this configuration, the combined diagonal pushing outflow (light blue arrows) and compensating diagonal inflow of equal magnitude (orange arrows) create parallel flows in opposite directions on either side of the stripe (red arrows). Hence, the result is a pure shear flow with invertible directions for ±45 degrees offset between the polarization and the stripe axis (Fig. 4g, h and Supplementary Movie 15). Under shear flow conditions, particles traverse the entire field of view in opposite directions alongside the rectilinear illumination stripe (Supplementary Fig. 17), a behavior that, once again, cannot be explained by the illumination pattern only. Instead, both the intensity pattern (here: illumination stripe) and the polarization (here: diagonal

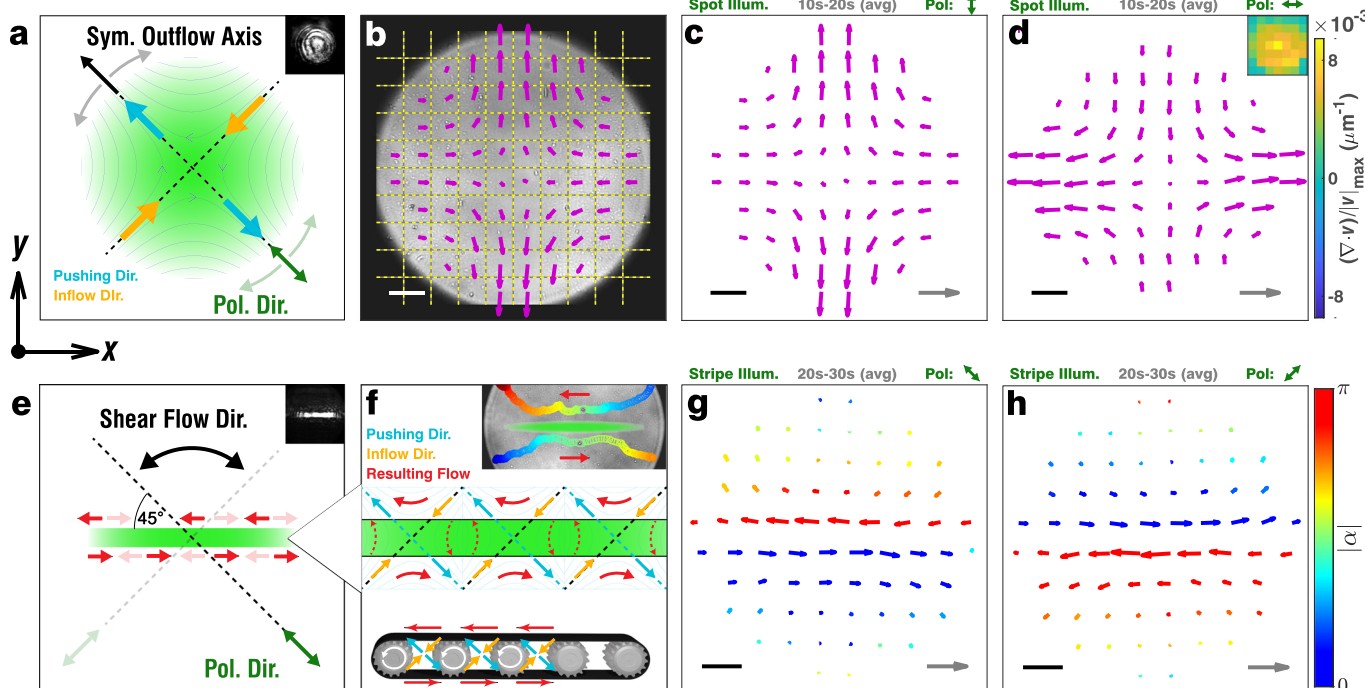

**Fig. 4 | Polarization-controlled rapid flows. a**) Scheme of adjustable polarization direction whose rotation will lead to a rotation of the entire flow-field at the water–air interface, including the symmetric outflow axis. Blue arrows: deformation/pushing direction. Orange arrows: inflow. **b** Microscope snapshot of Supplementary Movie 10, showing the cutting of the field of view into discrete tiles (black/yellow lines), with one computed flow-field vector (magenta) per tile (and time interval). Scale bar (white): 20 µm. **c**, **d** Flowfields associated to the time interval 10–20 s after illumination onset for y- and x-oriented linear polarization, respectively. Scale bar (black): 20 µm. Velocity reference arrow (gray): 20 µm/s. Inset: Relative divergence of the flowfield averaged over six consecutive flow fields. **e** Scheme of switchable shear flow obtained by tilting the polarization to 45 degrees

with respect to a stripe-shaped illumination. **f** Explanatory inset showing the origin of a pure shear flow for negligible local divergence (inflow = outflow). Blue arrows: pushing outflow. Orange arrows: compensating inflow. Red arrows: resulting (x-oriented) net flows. The conveyor analogy (bottom) shows how the transport movement of such a device may be decomposed into analogous components. The upper inset shows single tracked particles moving across the field in opposite directions along the illumination stripe, from blue to red (full frame in Supplementary Fig. 17). **g**, **h** Flowfields computed from data shown in Supplementary Movie 15, demonstrating such shear flows. Arrows are rainbow-colored with respect to the absolute value of their angular deviation from the x-axis. Scale bar (black): 20 µm. Velocity reference arrow (gray): 40 µm/s.

polarization) have to be considered to explain the overall shear flow pattern. Finally, at slightly increased densities, occasional turbulence with rotational motion was observed within the stripe region (Supplementary Movie 16), and the lateral shear flow velocities reached up to 90 µm/s (Supplementary Fig. 18), corresponding to ~100 body sizes per second for the particles.

## Discussion

In this work, we studied photo-deformable colloidal azopolymer particles dispersed at a water–air interface. In particular, we used the particles' optical deformation to create strong tunable capillary forces and to control their dynamic interactions directly at the interface. Employing linearly, elliptically and circularly polarized illumination profiles, we induced permanent anisotropic stretching and/or isotropic in-plane expansion of the particles, enabling their shape-mediated assembly and disassembly. Interestingly, continued directional deformation of capillary-assembled particles led to pushing of deformed particles out of, and inflow of fresh particles into, the static zone of illumination. Using this effect, we were able to drive sustained symmetric particle flows across the interface, whose direction we adjusted by changing the illumination's polarization. Furthermore, under appropriate illumination conditions, we could design in-plane shear flows with velocities up to 90 µm/s and with controlled directionality.

The first two steps in this cascade of dynamic, deformation-driven effects—photo-deformation of particles at interfaces and ensuing capillary control—may inspire the development of refined material systems for use in colloidal science. Monodisperse particles may be

developed, which could be assembled into monolayers before targeted optical deformation, enabling close-packed interface lattices incorporating non-spherical particles[24,44]. Controlled particle sizes may also facilitate the deterministic assembly and modification of more complex particle patterns. In this context, reversibly deforming particles may be a benefit (the particles in this work being limited to one assembly-disassembly cycle, after which they remain heavily flattened). Reversible photo-deformation of particles at the interface, allowing for repeated capillary assembly-disassembly cycles, could be achieved using light-responsive LCE particles[59] or recently introduced cross-linked amorphous azopolymers[36,60].

On the other hand, the sustained polarization-steered flows inscribe themselves in a growing context of strategies to control the individual and collective motion of particles and droplets in various media with light[61]. When directional control over such motion is envisaged, some symmetry breaking is generally required, which may be brought about e.g., by directional spatiotemporal intensity gradients, the choice of incidence direction, or polarization states. To mention some elegant examples from recent years, directional propulsion has been controlled by intensity gradients in one-dimensional capillary tubes[62], or by gradients combined with, e.g., Janus[63] or anisotropically buckling[64] particles in 2D systems. Tilted incidence illumination can make azobenzene microcrystals swim or crawl directionally[49,65] or induce particle-transporting self-shadowing waves[66]. Sweeping light patterns can directionally guide Marangoni flow traps[54,67] and create propulsive undulating features in particles[68]. Other approaches to directional transport include light-switch systems using molecular motors' helicity inversion[69], or spontaneous

symmetry breaking for optothermal capillary rafts[10]. Importantly, optically steered particles with distinct responses to different polarization states have also been developed, for example, in the case of light-responsive plasmonic microdrones[70,71].

Here, yet another symmetry-breaking mechanism based on illumination polarization was discovered, whereby the collective particle outflow direction directly follows the electric field orientation (polarization) of an otherwise axis-symmetric Gaussian beam. Since the symmetry-breaking in this system fundamentally arises from the polarization-directed particle photo-deformations—whereas the capillary forces act to collectively propagate these deformations into macroscopic directional flows—mechanisms and material systems not relying on interfacial capillary forces as conveyors can also be envisioned. Further, the combination of local polarization-dictated directionality with other means of directionality control, such as complex intensity patterns, was only briefly explored in the stripe illumination example. More elaborate configurations of intensity and polarization distributions may be explored, and in particular, the use of non-uniform polarization patterns considered[72]. Application-wise, the concept developed here appears promising for tasks such as drag-induced displacement and orientation of passive objects floating on liquid interfaces, for mixing in shear flow mode (with built-in transverse velocity components as in efficient microfluidic approaches)[73], or for flows driving the rotation of microgears[74].

## Method

### Particle fabrication
The polymer poly[(Disperse Red 1 methacrylate)-co-(methyl methacrylate) pDR1m-co-mma (Sigma-Aldrich, ~15 mol% dye monomer) was dissolved in chloroform at 10 | 25 mg/mL for samples shown in Supplementary Fig. 1a, b, respectively. 1 mL of solution was then added to 12.5 mL of deionized (DI) water containing 0.5 wt% polyvinyl pyrrolidone (Sigma-Aldrich, MW: 360 kD) | polyvinyl alcohol (Sigma-Aldrich, MW: 13–23 kD, 87–89%, hydrolyzed) respectively as stabilizer. The phase-separated mixtures were sonicated in an ultrasound bath (Elmasonic P30H, 37 kHz, 100%) for up to 5 h, interrupted by sporadic manual shaking only, until appearing homogeneously emulsified upon visual inspection. Subsequently, the chloroform solvent was evaporated by transferring the emulsions to an open beaker and sonicating further, whilst slowly raising the bath temperature to the boiling point of chloroform (~0.1 °C/min, from 55 °C upwards). After passing the boiling point, the samples were sonicated for 30 more minutes at 75 °C. Finally, the solutions were filtered with a 10-µm syringe filter, and excess surfactant was removed by three times centrifugating the particles and redispersing them in DI water.

### Water–air interface
Water–air interfaces were prepared inside dedicated cells, made from two glued microscopy glass sides, where the top slide figures a circular, laser-cut aperture (~5 mm in diameter). The cells were cleaned with a NaOH solution before each use and subsequently filled with 5 µL of DI water. After dilution to a suitable particle density, 10 wt% isopropyl alcohol was added as a spreading agent. Then, 0.5 µL of the final mixture were carefully deposited on the interface using a pipette. The sample was left to evaporate in open air for 5 min, before it was sealed by a dedicated transparent glass slide fixed with vacuum grease, such as to limit convective and evaporative flows and permit long enough observation times.

Predominant particle adsorption at the water/air interface was verified by confocal laser scanning microscopy (Zeiss LSM 980) using z-stack imaging with a 20×/0.8 NA objective, 514 nm excitation wavelength, and detection wavelength between 516–639 nm. To estimate the surface coverage for movies in sections 1–3, 20 raw frames were selected from the initial seconds of the movies, where no illumination is provided yet. These frames were then inverted, and subsequently, a particle

localization with a thresholding detection algorithm was applied. After locating the center of the particle, we obtained its area as the sum of all pixels surrounding the center that differ from the background intensity level and averaged over the considered frames. Examples are shown in Supplementary Fig. 19. Notably, the measured surface coverage is based solely on the area of particles large enough for detection and does not account for the surrounding steric stabilization layer[39]. The actual surface coverage is therefore expected to be substantially higher.

### Optical setup
The optical setup consisted of a regular inverted microscope, illuminated by an LED (Thorlabs MNWHL4) with a 10x objective as a condenser from the top and with a 40x/0.75 NA objective mounted below the sample for observation. For recordings, a CCD camera from IDS (UI-3060CP-M-GL R2) was used, which permitted reliable acquisition at fixed frame rates of 30 and 50 fps. A 532 nm wavelength continuous wave laser diode was employed for the particle deformations. The beam was introduced from below and focused into the vicinity of the back-focal plane for suitable expansion of the beam width. Intensity was adjusted by means of neutral density filters, and polarization was controlled with zero-order waveplates. For assembly experiments, a quarter waveplate's fast axis was oriented at 0, 22, and 45 degrees with respect to the axis of the incoming x-directed polarization, to obtain linear x-axis, elliptical (2.5:1 axis ratio) and circular polarization, respectively. To rotate the linear polarization in the flow experiments, a half-wave plate was used instead. Note that in Fig. 1d, the images (as well as the underlying movie) are rotated back by 22 degrees for visual alignment with the linear polarization case. Finally, to create the stripe-shaped profile in Fig. 4e–h, the neat sides of a scribed silicon wafer were mounted on a holder in parallel, facing each other, and inserted in a plane conjugate to the sample plane, leaving a small slit defining the stripe-shaped profile. The useful intensity range exploited in flow experiments was comprised between 26 and 645 W/cm², and intensities around 200 W/cm² were used in sparse particle experiments.

### Particle size characterization
To prepare samples for scanning electron microscopy (SEM), particles from both samples were diluted and dried on regular silicon wafer pieces. They were then sputter-coated with 10 nm of platinum-palladium. Large overview images were obtained using a standard-mode ETD detector (secondary electrons), 5 kV acceleration voltage and 0.1 nA electron current (image size 6144 × 4096 pixels, 33.7 nm/pixel). Particle outlines and sizes were detected in the program imageJ, using the following sequence: *Gaussian Blur (2.5px) → Auto Local Threshold v1.11 ("mean", 15px, P1 = −15) → Convert to Mask → Fill Holes → Watershed → Analyze Particles (Circularity: 0.6–1.0)* using 5|3 images from distinct regions for samples shown in Supplementary Fig. 1a and Supplementary Fig. 1b, respectively. Finally, the absolute particle counts were reweighed by the particle cross-section in 2D, using the following equation (which leaves the total amount of renormalized particle counts unaltered):

$$h_i = \frac{c_i \cdot n_i}{\sum_j c_j \cdot n_j} \sum_j n_j \quad , \qquad c_i = \pi \cdot r_i^2 \tag{1}$$

where $n_i$ designs the net particle count in the $i$th histogram interval, $h_i$ designs its reweighted value used for the histograms in Supplementary Fig. 1e, f (renormalized particle counts), and $r_i$ designs the average radius of the particles in the $i$th interval which is used to calculate the corresponding cross-section $c_i$. The distributions were plotted on a logarithmic scale.

### Three-phase contact angle by gel-trapping technique
The three-phase contact angle of the azopolymer particles was determined by the gel-trapping technique developed by Paunov[75].

49 mg agarose (ROTH) was dissolved in 3 mL water and heated to 60 °C. After equilibration for 30 min, 100 μL of the particle dispersion in 90:10 (v:v) water-isopropanol was spread at the air–water interface using a micropipette. Following an equilibration period of 1 h, the samples were allowed to cool to room temperature to enable agarose gelation, which occurred over ~2 h. Subsequently, polymethylsiloxane (PDMS; SYLGARD 184—Dow) prepared by mixing the base and curing agent at a 10:1 mass ratio was poured over the agarose gel. The PDMS was left to cure for 5 days. Once cured, the PDMS layer was carefully peeled off the gel. Residual agarose was removed by immersion in 90 °C water followed by rinsing. Finally, the PDMS films were air-dried.

The resulting PDMS films were analyzed using atomic force microscopy (AFM; Nanoscope Icon with closed-loop, Bruker) operated in tapping mode with OTESPA tips (MikroMasch; nominal resonance frequency: 300 kHz, spring constant: 26 N/m, tip radius: <7 nm). AFM images were processed using Gwyddion. All images were leveled to remove tilt, and the minimum height value was set to zero. From cross-sectional profiles of at least 20 particles, the height (H), and arc length (S) were extracted to calculate the three-phase contact angle (θ)[75].

### Particle detection at interfaces

Where individual particle outlines are fitted, e.g., to obtain particle areas and positions, dedicated procedures are applied in ImageJ and post-processed in MATLAB 2021. For example, to determine the aspect ratios and areas for the single particle deformation statistics in Supplementary Fig. 4, the following sequence was applied to entire stacks after cropping them to the relevant single particle area: *Auto Local Threshold v1.11 ("Phalsankar", 15px, P1 = 0.1, P2 = 0.2) → Convert to Mask → Median (1px) → Close → Fill Holes → Median (1px) → Analyze Particles (Circularity: 0-1, Area: 100px-Inf)*. From the detected particles, only the largest one, generally representing the particle of interest, was kept. When another particle entered the field of view towards the end of the sequence, the area was occasionally re-cropped for this part of the movie to the particle of interest. The obtained values were then post-processed in Matlab to remove noisy outliers due to faulty detection on single frames, by automatically rejecting unphysical results (with respect to the movies), consisting of detected aspect ratios above 6, as well as relative areas below 1 and above 8. Outliers with a distance to the surrounding mean higher than the surrounding standard deviation were also removed, and finally, all detected outliers were replaced with the surrounding mean.

Similar approaches relying on local thresholding were applied for the detection of initial particle positions in Supplementary Fig. 6 and for assessing individual particle areas in Supplementary Fig. 10, without further post-processing. The exact procedures are included, together with the segmented images and extracted values, in the shared data folder. To track two prominent particles in Supplementary Fig. 17 (also upper inset in Fig. 4f), an inverted global threshold was utilized.

### Flow analysis algorithm

The analysis code was written in MATLAB 2021. To analyze the flows, a total number of spatial tiles $n_x \cdot n_y$ and a temporal step length $\Delta T$, were defined. The movie stacks were then split into spatiotemporal sub-stacks, each covering the size of one tile in space ($\Delta x \cdot \Delta y = 1/n_x \cdot 1/n_y$) and spanning one timestep $\Delta T$ in time. The code was optimized for parallel computation of subsequent timesteps to reduce the overall computation time. Eventually, each sub-stack would lead to one velocity arrow, representing the average flow at the given tile position and for the time interval [$t_k$, $t_k + \Delta T$] where $t_k$ designates the onset time of the $k$th timestep. The procedure to extract this single flow arrow from its corresponding sub-stack is summarized below.

First, after subtraction of—and normalization by—the mean pixel value for each sub-stack image, the correlation between image pairs separated by a given number of frames $\Delta f$ was computed and averaged. This gives a temporal autocovariance function, which is decaying for increasing $\Delta f$, and which is similar to the decay function used in e.g., Dynamic Light Scattering experiments. In the spatiotemporal correlation maps detailed below, these values correspond to the decaying central pixel. Next, all the images of the sub-stack would be shifted by -$i$ pixels in the $x$-direction and -$j$ pixels in the $y$-direction, constructing a second stack. The same correlation would then be computed for image pairs where the first image would be taken from the initial non-shifted stack, whilst the image delayed by $\Delta f$ frames would be taken from the shifted, second one. Note that this distinction of the delayed image always being the shifted one, is crucial to avoid temporally symmetric results and to be able to extract drift directions as schematically depicted in Supplementary Fig. 20. Finally, for each value of $i$, $j$, and $\Delta f$ the obtained correlation value was plotted as the pixel in position ($i$,$j$) with respect to the center on the $\Delta f$-th correlation map. Examples of the obtained correlation maps, which depict the full, averaged, spatiotemporal drift-diffusion behavior of the particles pertaining to the sub-stack, are shown in Supplementary Fig. 21.

We note that the 0th correlation map (containing the correlation between image frames with shifted versions of themselves) simply shows the spatial autocovariance, which may be anisotropic, for example, if the particles in this part of the field of view and at the given time on average are already anisotropically deformed. The $\Delta f$-th correlation map then shows the probability of image features (e.g., particle outlines) moving to new relative positions after $\Delta f$ frames, on average. This spatiotemporal evolution may be influenced by diffusion, drift (with velocity dispersion), and deformation of features. Here, the only aim was to extract the average drift velocity, which was performed by tracking the average movement of the center of the spatiotemporal probability distribution. Hence, for each correlation map linked to a $\Delta f$ delay, we started from the pixel with the highest correlation value and attempted to fit the exact peak position within this pixel by a 2D Gaussian fit of the whole map. Finally, the obtained peak positions were plotted against $\Delta f$, and the average drift velocity was obtained by a linear fit of motion (see Supplementary Fig. 21, bottom), followed by conversion into the final units (from pixel/frame to μm/s). The resulting arrow was then plotted, alongside the arrows from the other tiles, on the flowfield pertaining to the timestep $t_k$. In Fig. 3d, we plot only the $y$-component of the velocity for the tile of interest.

To calculate divergence maps of entire flow fields, origins were placed in each center between four flow arrows. The area used to compute an approximation to the local divergence was chosen as a square at an angle of 45 degrees with the velocity vectors' grid (see Supplementary Fig. 22). Then, the total outflow of the four vectors via each side of the square, divided by the total square area, was calculated to obtain the local average divergence. Note that horizontally/vertically adjacent tiles partially overlap (acting as a moving average) and that diagonally adjacent divergence tiles are not independent, since they always share one arrow. Hence, when taking the mean divergence of the whole field, as done in Supplementary Fig. 15f, in practice, one computes the outflow only through the total outer boundary of the considered tile region, since internal arrow contributions will cancel out.

## Data availability

Raw data were made available on the Figshare repository https://doi.org/10.6084/m9.figshare.28067498. Specifically, the original flow movies, with full frame rate and resolution, are included, as well as raw data concerning single particle deformations and capillary interactions. Any data or relevant information is available from the corresponding authors upon request.

## Code availability

The annotated correlation analysis code, and a list of code parameters used for the analysis is made available on the Figshare repository https://doi.org/10.6084/m9.figshare.28067498.

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

## Acknowledgements

D.U. acknowledges the Norwegian University of Science and Technology for funding the PhD fellowship under project number 989454111, and the Research Council of Norway is acknowledged for the support to the Norwegian Micro- and Nano-Fabrication Facility, NorFab, project number 295864. M.R. acknowledges funding from Sklodowska-Curie Individual Fellowship (Grant No.101064381) and Deutsche Forschungsgemeinschaft (DFG, German Research Foundation)—SFB 1459/2 2025—433682494. E.D. acknowledges the project "Metastatic potential of cancer cell aggregates in deformable matrices controlled by light"— CUP E53D23015220001, funded by the Italian Ministry of University and Research under the National Recovery and Resilience Plan (NRRP), Mission 4, Component 2, Investment 1.1, Call for tender No. 1409. G.V. acknowledges support from the project ERC CoG project MAPEI sponsored by the European Commission (Horizon 2020, Project No. 101001267) and from the Knut and Alice Wallenberg Foundation (Grant No. 2019.0079). We thank Giuseppe Leonetti (National Metrology Institute of Italy) for technical assistance. We also thank Thomas Zobel and Sarah Weischer of the MÜNSTER IMAGING NETWORK of the University of Münster for support in confocal microscopy. DFG-Antragsnummer: INST 211/898-1 FUGB. The authors would like to acknowledge the Geilo School 2022 on the physics of evolving matter, which provided a fertile ground for concretizing the interdisciplinary ideas underlying the present work.

## Author contributions

D.U., M.R., G.V., and E.D. conceived and planned the work jointly. D.U. prepared the samples. D.U., M.R., and A.C. performed the experiments, under the supervision of G.V. and E.D. D.U. performed data analysis, visualization and illustrations. A.C. evaluated surface coverage and MFS measured contact angles. All authors contributed to discussing and interpreting the results, as well as to the writing and reviewing of the manuscript.

## Competing interests

The authors declare no competing interests.
