## [Transparent Peer Review file · Nature Communications]

Directional flows using capillary assembly of photo-deformable colloidal particles at water-air interfaces

Corresponding Author: Professor Emiliano Descrovi

Version 0:

Reviewer comments:

Reviewer #1

(Remarks to the Author)

The manuscript deals with self-assembly of deformable particles at liquid interfaces, whose shape can be modulated. Authors have reported a simple method to modulate the shape of particles at the interface, studied their self-assembly at fluid interfaces and comment about interesting interfacial flows that can be generated. Overall, I see several conceptual issues and lack of new insight about interfacial assembly of colloids. I am of the opinion that the manuscript is not a adequate quality of publication in Nat Com. Following are major concerns:

(1) The results presented in the section "Optically controllable capillary interactions and assembly" are not new. There have been several studies where similar interfacial arrangement or self-assembly of particles has been reported.

(2) The shape modulation that is claimed is very limited. There are several methods for the controlled synthesis of non-spherical particles, including those of different elongation. Authors do neither demonstrate versatility of shape transformation nor the ability to create unique structures at interfaces.

(3) Figure 3: Quality of the microscopy images is very poor. Its hard to believe if the particles are at the interface or they are in the bulk.

(4) Authors discuss experiments at higher density where interfacial flows are generated, but there is no quantification of surface coverage, and the lack of flow in dilute systems.

Reviewer #2

(Remarks to the Author)

I have reviewed the manuscript NCOMMS-24-84692 titled "Directional capillary flows induced by photo-deformable colloidal particles," authored by David Urban et al.. Their research focused on morphological alterations and subsequent aggregation or separation of amorphous azobenzene polymer particles dispersed on aqueous surfaces. This study revealed that when exposed to linearly polarized visible light, these azobenzene particles exhibited continuous movement along the polarization direction. This phenomenon is likely to captivate readers' interest. However, the authors' discussion that this continuous flow resulted from capillary forces generated by rod-shaped particles lacks sufficient supporting evidence. This manuscript requires additional experimental data and thorough analysis. Furthermore, there is a lack of literature research on the light-driven locomotion of nanoobjects on water surfaces. The detailed comments are as follows.

1. To establish the novelty of this manuscript, the authors should extensively review the existing literature on the movement of powdered substances or particles on water surfaces through irradiation as well as other forms of nano-locomotion. This comprehensive review should include, but is not limited to, studies such as Norikane et al. *Cryst. Eng. Commun.* 2016, 18, 7225; Fujii et al. *Adv. Funct. Mater.* 2016, 26, 3199; and Lucchetta et al. *AIP Adv.* 2015, 5, 077147.

2. What method did the authors employ to achieve stable dispersion of individual azobenzene polymer particles at the water-air interface? Typically, particles tend to aggregate on water surfaces or disperse in the water. Successful stable dispersion of azobenzene polymer particles is a key technique in this research. Consequently, the authors should provide a detailed

explanation of their technique for creating a stable dispersion on the water surface, supported by experimental evidence, such as water contact angle measurements.

3. The authors should discuss the photoisomerization of the azobenzene polymer particles more carefully.

3.1. The authors should clarify how the laser wavelength correlates with the absorption of the azobenzene polymer. They must also include the time evolution in the UV-vis absorption spectra of the azobenzene polymer before, during, and after laser exposure. This information is crucial for analyzing the light-induced properties of azobenzene-based materials and represents one of the most fundamental datasets in this study.

3.2. More quantitative analyses of the particle morphology changes and movements are necessary. The wavelength and intensity of the light source typically influence the light-activation properties of azobenzene-containing materials.

3.3. The factors affecting capillary forces should be discussed more carefully. The surface free energy, as well as the shape of the particles, influence capillary forces. During $n-\pi^*$ absorption, azobenzene is known to undergo trans-cis conformational interconversion. This photoisomerization process results in two notable effects, in addition to shape alterations in azobenzene materials. First, the photothermal effect occurs, where irradiation elevates the temperature of azobenzene materials. Second, there is an increase in the composition of cis-isomers, which exhibit greater hydrophilicity than trans-isomers. Both temperature changes and increased hydrophilicity can alter the surface free energy of azobenzene polymer particles. These phenomena should be addressed in the Results and Discussion sections. For instance, when a linearly polarized 532-nm laser irradiates one side of the azobenzene polymer particles, it can create a gradual difference in the surface free energy across the particle surface, resulting in nano-locomotion on water surfaces.

3.4. To substantiate the claim of "shape-mediated assembly and disassembly" as discussed in the Discussion section, where the authors claimed that capillary forces resulting from rod shapes would be the sole cause of continuous locomotion, they must thoroughly rule out alternative explanations. This includes eliminating the possibility of other effects, such as Marangoni propulsion, which has been extensively researched in the field of nano-locomotion. The authors should address this issue using appropriate scientific methodologies.

3.5. The authors should distinctly differentiate between nano-locomotion, which results from microscopic forces near particle surfaces, and macroscopic water movement. The initial part of the Results section noted that the azobenzene polymer particles aggregated together after changing from spherical to rod-shaped, even after the laser was turned off. This indicates that the capillary forces from the rod shapes promote aggregation rather than repulsion. Figure 3iii and Movie M7 provide clear evidence of rod-shaped azobenzene polymer particle aggregation even during irradiation. Concurrently, the clustered particles were drawn away from the irradiated area on the water surface. The authors suggested that water flow caused by osmotic pressure was a potential cause, inappropriately citing photoswitching of the adsorption/desorption behavior of azobenzene surfactants. They claimed that photoisomerization of azobenzene components increased hydrophilicity, creating the "sustained pumping machinery." However, this effect is highly unlikely in the current system, where no azobenzene derivatives are dissolved in the aqueous phase or detached from the particles; therefore, a local concentration of cis-isomers and the resulting osmotic pressure should not be generated. Consequently, this study lacks a compelling explanation for the continuous flow observed during irradiation.

3.6. As illustrated in Figure 3 and Movie M7, the nano-locomotion effect was the most pronounced in the central region of the scope. This observation suggests that the intensity of the laser progressively decreased from the center to the periphery of the exposed area. Such a gradient in laser power could also explain the observed continuous flow.

3.7. Additionally, when the light-induced motion of azobenzene materials typically ceases upon turning off the light source, the photothermal effect of azobenzene groups during illumination is generally considered the primary cause of this movement. Although hydrophilic cis-isomers are transformed back into hydrophobic trans-isomers under dark conditions, this conversion does not occur instantaneously after the light source is deactivated. As previously mentioned, the time-dependent changes in UV-vis absorption spectra would likely support this gradual conformational reversal process.

3.8. The rod-shaped particles aggregated after changing their shape, regardless of the irradiation. This suggests that deformed particles expelled from the irradiated area would aggregate together once the laser was turned off. However, Movie 7 demonstrates that the deformed particles remained stationary on the surface of the water after laser deactivation. This raises the question of what accounts for this discrepancy.

Reviewer #3

(Remarks to the Author)

The manuscript entitled "Directional capillary flows induced by photo-deformable colloidal particles" by Urban et al. presented very interesting assembly and disassembly of azopolymer-based shape-deformable microparticles at the air/water interface in response to polarized light. However, to precisely evaluate this work, the capillary flow-driven mechanism should be clarified.

(1) The authors applied highly intensive polarized light to drive the shape deformation. In addition to photoisomerization, azobenzene-based aggregates would perform photothermal conversion under light irradiation, which enables driving directional movement at the air/water interface as well (e.g., CrystEngComm, 2016, 18, 7225). The authors should clarify whether there is a photothermal behavior or not. And the impact of this behavior on the directional assembly and particle

flow.

(2) Similarly, such highly intensive light is another possible source producing a thermal effect on the water, which may drive particle flow. The authors should clarify its impact on the water and strong photoabsorbing azopolymer particles.

(3) Photoisomerization is another possible driving force for directional movement (e.g., *Advanced Functional Materials*, 2016, 26, 3164). Because a change of polarized light would lead to assembly and disassembly, the authors should clarify the impact of photoisomerization on directional particle assembly.

(4) Considering the capillary effect, the mechanism of assembly and disassembly is still not very clear. The particle shape is very important in capillary assembly, but lots of research shows the spheres can also achieve capillary assembly. Why does the shape deformation here lead to a rapid disassembly? Is there an impact of interfacial interaction on stabilizing or destabilizing particle aggregates? Does changing the light from elliptical polarization to linear one or reverse operation have some impact on the assembly or disassembly? If the explanation is the variation of out-of-plane curvature radius as the authors claimed, the particle dimension should play an important role in the capillary-driven assembly and disassembly. The authors should provide information on this impact.

(5) The effective length (or interparticle distance) of the capillary effect in driving directional particle assembly should be clarified.

Version 1:

Reviewer comments:

Reviewer #2

(Remarks to the Author)

The revised manuscript has been enhanced and is now much clearer than the original version. The authors have meticulously addressed my comments one by one. This version effectively clarifies that the continuous flow was not a direct result of traditional nano-locomotion due to photoisomerization, but rather from the ongoing self-assembly of ellipsoids when exposed to a linearly polarized light. These self-assembling movements then consistently expelled water in the same direction as the long axis of the self-assembled clusters. The concept of a novel driving force for particle flows proposed in this study may engage readers.

However, concerns have emerged regarding the adaptability of this strategy due to several limitations. Furthermore, the discussion in the concluding paragraph lacks persuasiveness in its assertions about the strategy's impact.

First, the functionality of the particle is notably constrained. The range of applicable materials is predominantly confined to DR1 polymers, characterized by a substantial overlap of the trans-cis bands. In the absence of this overlap, conventional nano-locomotion mechanisms are likely to be activated. This limitation in material applicability restricts the functional diversity of the particles. Additionally, the functional modification of particle surfaces is impeded because the balance between hydrophilicity and hydrophobicity on the particle surfaces is vital for achieving a dense distribution on the water surface while simultaneously preventing the initial formation of aggregates. The functional limitation constrains the potential application of this strategy.

Second, the lack of correlation with light intensity, which was stated in the rebuttal, hinders the precise quantitative control of directional flows. Although the authors suggest the potential application of various light properties, controlling directional propulsion by intensity gradients remains unlikely.

Moreover, the nonspherical shapes of the particles are insufficient for promoting diverse particle flows, as the generation of capillary forces requires one-dimensional deformation of the particles, irrespective of their initial shape. Nonetheless, the authors have proposed the potential application of buckling particles. Additionally, the use of Janus particles would interfere with the one-dimensional stretching of the particles.

The findings of this study must demonstrate substantial potential for future advancements that surpass the objectives delineated by the authors in the Discussion section.

The following are additional minor comments:

1. Although stretched particles predominantly self-assemble in a side-by-side manner, the formation of end-to-end configurations is also essential for the development of elongated assemblies. The end-to-end configurations are clearly illustrated in Figure 3c, panels iii and iv, along with the side-by-side configurations. The mechanism underlying the formation of the end-to-end configurations must be elucidated.

2. The head-tail configuration of the DR1 main chain structure depicted in Figure 1 is inverted. The sequence -C(CH₃)-CH₂- needs to be altered to -CH₂-C(CH₃)-.

3. The section headings should be revised. Most of the discussion is detailed in the Results section, while the Discussion section primarily summarizes this manuscript.

4. Considering the title and the focus outlined in the Introduction, it is inappropriate to conclude this manuscript by emphasizing the relevance of tilted incidence illumination to conventional nano-locomotion systems in the closing paragraph.

Reviewer #3

(Remarks to the Author)

The reviewer thanks the careful response and additional experiments for the revision. The response answers some questions. However, some content still sounds confusing.

1. Photoisomerization has shown a crucial effect on particle deformation. And photothermal conversion would influence the surrounding flow. The authors did not do the characterization. The UV-vis spectra and photothermal curves should be given to reflect photo-responsive behaviors.

2. If the reviewer understands correctly, the authors insist that there is a similar photothermal effect and photoisomerization in response to different polarization lights, so they would not give different influences on the flow. The change of polarization light would induce particle deformation into different shapes. The photothermal conversion within different-shaped particles would significantly influence the surrounding temperature distribution and flow direction (e.g., Lab Chip, 2012, 12, 3707). The authors should clarify that on particle locomotion at the interface.

3. Because the particle shape would influence the surrounding flow direction, the particle assembly into aggregates may further intensify this effect on the flow, leading to directional flow. The authors should clarify that.

In addition, I have also looked over the responses to the comments from Reviewer 1.

4. I agree with Reviewer 1. The current microscopy images cannot accurately localize the particles at the interface. And the authors did not give direct evidence for that in the response. It is better to use another characterization instrument (e.g., confocal microscopy with z-stacking scan) to give the particle distribution in the solution. That would also rule out the possible influence of the thermo-flow induced by the dispersed particle underwater.

5. The authors discussed the influence of surface coverage on particle assembly and disassembly in the revision. According to Reviewer 1's comments, the flow phenomenon of these dilute systems should be explored. Normally, there should be a thermal Marangoni effect to propel photothermal particles at the interface. In terms of high-intensity light used in this study, it is very surprising that the authors' response indicated there was no flow observed for the diluted systems. The authors should explain this phenomenon. That would be a good control in this study.

Version 2:

Reviewer comments:

Reviewer #2

(Remarks to the Author)

The authors have comprehensively addressed all my concerns, thereby enhancing both the manuscript and the Supporting Information. They showed that their findings regarding particle flows are not attributable to conventional mechanisms but rather to the deformation of particles, which induces capillary forces. Compelling data and discussions support this explanation. This approach holds significant promise for advancing novel technologies for photomechanical energy conversion. I recommend that this manuscript be suitable for publication.

Reviewer #3

(Remarks to the Author)

The authors have addressed the reviewer's concern. This version of the manuscript is recommended for publication.

We thank the reviewers for their detailed comments. In particular, we observed that several reviewers raised overlapping questions regarding the fundamental working mechanism of the flow phenomenon, the influence of azobenzene isomerization or thermal effects. In response, we revised the manuscript thoroughly by providing further experimental data, relevant literature, as well as an extensive discussion clarifying the proposed flow mechanisms and the intended message of the work. We hope that these clarifications address the reviewers' concerns and help to convey the significance and originality of our findings more clearly. For simplicity in referring to manuscript modifications, we refer to the result sub-sections in the following way:

"Section 1" for "Shape-morphing particles at an air-water interface"

"Section 2" for "Optically controllable capillary interactions and assembly"

"Section 3" for "Continuously deforming capillary assemblies"

"Section 4" for "Polarization-controlled directionality and shear flows"

Supplementary Figures and Movies are numbered with respect to their appearance in the revised manuscript and the attribute "new" Supplementary Figure/Supplementary Movie labels items that were not yet included in the initial submission.

Reviewer #1 (Remarks to the Author):

The manuscript deals with self-assembly of deformable particles at liquid interfaces, whose shape can be modulated. Authors have reported a simple method to modulate the shape of particles at the interface, studied their self-assembly at fluid interfaces and comment about interesting interfacial flows that can be generated. Overall, I see several conceptual issues and lack of new insight about interfacial assembly of colloids. I am of the opinion that the manuscript is not a adequate quality of publication in Nat Com. Following are major concerns:

General Comments: We thank the reviewer for providing their direct perspective on the work. From the feedback, we realize that we did not formulate the message of the article clearly enough. In particular, regarding the shape-mediated assembly of particles, we consider as novel not the assembly process itself, but the possibility to trigger and modify such assemblies locally and on demand with light, while the particles are already dispersed at the interface. Instead of achieving static assemblies, the capillary forces can thus be controlled during the actual experiment, permitting novel functionalities such as the polarization-directed flows that constitute the major effect reported in our opinion.

(1) The results presented in the section "Optically controllable capillary interactions and assembly" are not new. There have been several studies where similar interfacial arrangement or self-assembly of particles has been reported.

As mentioned above, we realize that the novelty of this part needs to be formulated more clearly. We are aware of the various shape-mediated assemblies that have been reported previously (References 11–14, 19–22 and 38 in the initially submitted draft). In the cited works, the particles were typically synthesized with a defined anisotropic shape—such as hematite or silica ellipsoids—or produced by mechanically stretching polymer spheres. As outlined in the Introduction section, once fabricated, the particle shape remained fixed, and self-assembly could only be explored for particles with globally predefined shapes and interactions.

From our perspective, the purpose of section two is to provide illustrative examples of the possibility to induce known shape-mediated capillary forces between colloidal particles *optically at the interface (this part is novel)*, to verify that these capillary forces are present in our system and simultaneously to introduce such capillary forces, which we find foundational to understanding the main phenomenon reported (polarization-directed

flows), to the readership outside of the pure colloidal interface field (e.g., light-responsive polymers, particle manipulation, microfluidics and adjacent fields).

Text modifications: To avoid similar misunderstandings about what the novelty of part 1 and part 2 is for potential readers of the manuscript. Specifically, we have rephrased several key sentences to better emphasize the conceptual distinction and innovation introduced in each part. The following examples illustrate these changes: -“...using capillary assembly of photo-deformable particles..”

“... we selectively trigger capillary self-assembly by anisotropically deforming the particles at the interface”

-“... shape-mediated, capillary-driven self-assembly—previously achieved using anisotropic particles that were fabricated in advance and then placed at liquid interfaces^{11–14,19–22,38}—can instead be induced in situ via light-driven deformation of initially spherical particles directly at the interface.”

(2) The shape modulation that is claimed is very limited. There are several methods for the controlled synthesis of non-spherical particles, including those of different elongation. Authors do neither demonstrate versatility of shape transformation nor the ability to create unique structures at interfaces.

We agree with the reviewer that we neither can, nor claim to produce particles with shapes that cannot be obtained in a more precise manner by *a priori* synthesis, i.e., before dispersion at an interface. For example, sophisticated mechanical stretching methods have been shown to offer control over all three ellipsoid axes of polymer particles, as well as possibilities to fabricate other shapes, in a highly reproducible and large-scale manner (Champion, J. A. et al., *Proceedings of the National Academy of Sciences* 104, 11901–11904 (2007), reference 17 in initial draft). Obtaining or demonstrating similar precision in the proposed interface-dispersed particle system is limited by the following factors:

- 1) From a technical perspective, obtaining a precise narrow shape distribution for all particles is limited (in this system) by the fact that we start with a polydisperse particle set (i.e., particles of broadly dispersed initial sizes see Supplementary Figure S1). Particle size is known to influence the deformation process in azopolymers, even for particles on dry substrates (see e.g., Loebner, S. et al., *J. Phys. Chem. B* 122, 2001–2009 (2018) or Liu, J. et al., *Langmuir*, 25 (10), 5974–5979 (2009) or Ryabchun, A. et al., *Advanced Optical Materials* 7, 1901486 (2019))
- 2) From a more fundamental perspective, it should be noted that the particles dispersed at the interface undergo (rotational) diffusion. Diffusion being a random process, this is expected to widen the distribution of shapes obtained under continuous deformation conditions in the experiment. We believe that this is an intrinsic limitation of the presented approach, at least with respect to narrow shape control in a fabrication perspective.

Text modifications: To show and quantify the shape modulations that can be obtained in the proposed system and answer the reviewer’s request, as well as to highlight limitations, we analyzed the (in-plane) deformation of 20 single particles each under illumination with linearly, elliptically and circularly polarized light and included the resulting graphs as a new Supplementary Figure S3 (for a detailed description please consider also the answer to reviewer 2’s question 3.2). The Supplementary Figure is referenced and discussed in the manuscript at the end of section 1. Whilst a considerable distribution width is present due to the factors described above, clear deformation trends can be highlighted. First the deformations are continuous, which allows for control over the average deformation extent through exposure time. Secondly, the in-plane anisotropic component of the deformations can be tuned by the polarization from linear (anisotropic stretching and flattening) to circular (in-plane isotropic flattening only) in a reproducible manner. Both the in-plane aspect ratio and the flattening being relevant for capillary interaction, this provides two degrees of freedom in the obtained average shapes.

Comment: The difference with respect to conventional approaches is that here the shape of individual particles may be addressed in a targeted way at the interface and may be altered several times, since it is not fixed during particle fabrication. Thus, we dare to hope that the present approach may be inspiring to some researchers in

the self-assembly field (note the Discussion part where approaches to obtain suitable particle systems for self-assembly are outlined). Still, the main focus in this work is not on self-assembly of particles into exotic structures, but on the exploitation of the dynamic features emerging from *in situ* shape modulation and from the associated capillary forces. We consider the polarization-directed dynamic flow phenomenon, which constitutes the main body of the work, an intriguing example of this possibility.

(3) Figure 3: Quality of the microscopy images is very poor. It's hard to believe if the particles are at the interface or they are in the bulk.

All experiments were conducted at a well-defined air–water interface. We kindly disagree that the image quality is poor and several lines of evidence confirm the interfacial confinement of the particles:

1. Imaging setup and focal plane control: All images and movies were acquired using a regular brightfield transmission microscope. The sample chamber consists of a sealed water–air interface confined between two glass slides (see Methods: "Water–air interface"), which minimizes evaporation and convective flows. To ensure interface stability and prevent dewetting, a minimum water height of 150 μm is maintained. This requires an objective with a sufficiently large working distance; we therefore use a 40 \times , 0.75 NA air objective as the optimal compromise between resolution and working distance. The corresponding depth of field is approximately 1 μm , i.e., comparable to the particle size. Particles that are simultaneously in focus can thus be considered to lie in the same horizontal plane.

We realize that it may be more difficult for the reader to verify that the particles are at the air-water interface than for the experimenter, who can scan through the vertical plane—from the bottom glass slide upward—to identify the focal plane where most particles lie, above which no particles are observed. This is particularly relevant in the sparsely populated examples shown in sections 1 and 2.

However, a brief look at the initial seconds of Supplementary Movie M9 (denser case) clearly demonstrates that the particles are confined to the interface: all particles exhibit lateral Brownian motion in the same focal plane (1 μm depth), without drifting in or out of focus. The same is true for movies with sparsely distributed particles, spanning durations of minutes. If the particles were dispersed in the bulk, their Brownian motion would cause them to move out of the focal plane within seconds. The consistent in-plane visibility thus clearly demonstrates that the particles are confined at the liquid interface.

2. Optical properties of the particles and image contrast: The particles contain an azobenzene dye, providing strong absorption in the blue/green spectral range, but are nearly transparent for long wavelengths (a UV-VIS spectrum can be found e.g., in Urban, D. et al., *Nat Commun* 14, 6843 (2023), in Supplementary Figure S1b). Their refractive index is up to ~ 1.7 in the red (Rodriguez, V. et al., *J. Phys. Chem. B* 107, 9736–9743 (2003)), compared to ~ 1.3 for water and ~ 1.0 for air. This provides good contrast for the initial spherical particles.

However, as the particles flatten under illumination, both the optical path differences and the attenuation decrease. This leads to significantly reduced image contrast for highly deformed particles. To partially compensate for this, we reduced the illumination aperture using an iris, which increases contrast at the expense of some spatial resolution. Despite contrast loss, sharp features can still be appreciated on heavily flattened particles (for example in Figure 1c, right), implying that they still are in the same focal plane.

3. Rotation diffusion of anisotropic particles reveals that they must be confined to the liquid interface:

Additional evidence comes from the rotational diffusion of anisotropic (deformed) particles, such as elongated rods or flattened disks. Elongated rods undergo rotational diffusion, yet their long axis remains within the imaging plane (perpendicular to the optical axis). Similarly, flattened disks always present their broad face to the observer. This clearly indicates the absence of three-dimensional rotational diffusion as would be expected for particles moving in bulk water. Further, it is consistent with non-spherical particles commonly being adsorbed at interfaces

with orientations minimizing the unoccupied interfacial area, i.e., with the largest cross-section parallel to the interface (Botto, L., et al., *Soft Matter* 8, 9957–9971 (2012)).

Text modifications: To remove all doubt about the particles being dispersed at the interface and provide additional information as requested by reviewer 2, we performed additional experiments measuring the three-phase contact angle of the particles by gel trapping, included in New Supplementary Fig. S2. In addition, we have improved a sentence in section 1, clarifying the reasons for contrast loss in heavily flattened particles to the reader.

(4) Authors discuss experiments at higher density where interfacial flows are generated, but there is no quantification of surface coverage, and the lack of flow in dilute systems.

We agree that estimations of surface coverage is helpful for guiding researchers in reproducing the results. We can estimate the surface coverage for the larger particle set by detecting particles directly from the optical microscope images of the movies (see new Supplementary Figure S17 below), on frames selected from the interval before illumination. The values (mean \pm s.d. over 20 frames) obtained from the different movies (full field of view) are:

Supplementary Movie M2, two-particle interaction = (0.74 ± 0.09) %

Supplementary Movie M3, multi-particle interaction: (0.73 ± 0.02) %

Supplementary Movie M4, two-particle interaction: (0.69 ± 0.03) %

Supplementary Movie M5, multi-particle interaction: (0.63 ± 0.10) %

Supplementary Movie M7 (first part), multi-particle interaction: (2.07 ± 0.07) %

Supplementary Movie M7 (second part), multi-particle interaction: (1.97 ± 0.18) %

Supplementary Movie M8 (second part), multi-particle disassembly: (2.09 ± 0.02) %

Supplementary Movie M9, flows: **(38.89 ± 0.25) %**.

We therefore summarize that the surface coverage is <5% for the experiments pertaining to the opto-capillary assembly and disassembly part (section 2). The surface coverage for the flow regime shown in section 3 is 38%. Note that the smallest particles in the distribution may not accurately be counted but represent a manageable surface coverage fraction compared to larger particles, as shown in Supplementary Fig. S1e.

The reviewer is correct in that no flows are observed in dilute systems. This is because the flow phenomenon reported is not a single particle effect (such as e.g., thermophoresis), but an effect arising from the collective assembly and deformation of dense particles, as outlined in more detail in the answer to Reviewer 2's question 1.

New Supplementary Figure S17. Surface Coverage. The surface coverage was evaluated for movies in section 1-3 based on the following approach. First, 20 frames were selected from the initial seconds of the movies, where no illumination is provided yet (example images on the left). These frames were then inverted (example images, middle) after selecting the circular region of interest (red circle) and subsequently a particle localization with thresholding detection algorithm was applied. After locating the center of the particle, we obtained its area as the sum of all pixels surrounding the center that differ from the background intensity level (example images, right). The total fraction of the area occupied by detected particles was averaged over 20 frames, with mean values and standard deviation indicated to the right. a-c) Example images pertaining to Supplementary Movie M9, showing flow development described in section 3 (Figure 3). d-f) Example images pertaining to Supplementary Movie M7 (second part), showing an assembly-disassembly cycle involving multiple particles, described in section 2 (Supplementary Fig. S8b). g-h) Example images pertaining to Supplementary Movie M2, showing an assembly-disassembly cycle involving two particles, described in section 2 (Figure 2a). Coverage is significantly lower in all movies pertaining to the assembly-disassembly experiments than in movies used to assess the flow phenomenon.

Text modifications: We added the general surface coverage indications indicated above to the manuscript at the beginning of section 3 and the method for estimation of the quantity to the Method section *Water-Air Interface*.

Reviewer #2 (Remarks to the Author):

I have reviewed the manuscript NCOMMS-24-84692 titled "Directional capillary flows induced by photo-deformable colloidal particles," authored by David Urban et al.. Their research focused on morphological alterations and subsequent aggregation or separation of amorphous azobenzene polymer particles dispersed on aqueous surfaces. This study revealed that when exposed to linearly polarized visible light, these azobenzene particles exhibited continuous movement along the polarization direction. This phenomenon is likely to captivate readers' interest. However, the authors' discussion that this continuous flow resulted from capillary forces generated by rod-shaped particles lacks sufficient supporting evidence. This manuscript requires additional experimental data and thorough analysis. Furthermore, there is a lack of literature research on the light-driven locomotion of nanoobjects on water surfaces. The detailed comments are as follows.

General comments: We thank the reviewer for his well-structured question list. The questions mainly concern the physical origin of the directional flow phenomenon reported. This phenomenon was discovered unexpectedly by the authors and is fairly exotic, involving effects quite different from those conventionally used to cause the locomotion of particles/objects at liquid interfaces. Thus, before addressing the questions in detail, we would like to point out the following general points:

- 1) The flow phenomenon is a collective effect which does not affect single, sparsely distributed particles such as shown in the first two sections. In fact, it is not even a locomotion effect for freely diffusing particles per se, since it consists in the deformation of large (static, in the absence of light) capillary assemblies of particles as a whole, i.e. of capillarity-assembled particles propagating their light-induced deformation.
- 2) We are aware that most works involving azobenzene molecules in the literature employ the targeted *trans* → *cis* or *cis* → *trans* conversion of molecules (often using distinct wavelengths to selectively favor one of the conversions). The molecule used here (Disperse Red 1) is a pseudo-stilbene, where the *trans* and *cis* bands overlap heavily, and where one often exploits continuous *trans* → *cis* → *trans* cycling at a single wavelength to cause statistical dye reorientation (for comprehensive reviews, see e.g., Natansohn, A. & Rochon, P., *Chem. Rev.* 102, 4139–4176 (2002), focusing heavily on dyes as employed here or Mahimwalla, Z. et al., *Polym. Bull.* 69, 967–1006 (2012), comparing the different classes of azobenzene molecules). Eventually polarization-sensitive deformation effects in the polymers ensue from reorientation and cycling (see e.g., the reviews Saphiannikova, M. et al., *Soft Matter* 20, 2688–2710 (2024) and Priimagi, A. & Shevchenko, A. *Journal of Polymer Science Part B: Polymer Physics* 52, 163–182 (2014).).
- 3) All experiments shown in this work are under illumination incidence normal to the liquid interface and produce strongly polarization-dependent results (both regarding assembly-disassembly and flows). Neither *trans-cis* isomer populations nor photo-thermal absorption effects are strongly polarization-dependent. On the other hand, the photo-deformation of particles, which we believe to be the driver of the flow phenomenon (not capillary forces alone), is strongly polarization-dependent and can account for the flow symmetry breaking observed.

In the following, we substantiate these explanations in detail and address the detailed questions with the help of additional experiments and links to existing literature.

1. To establish the novelty of this manuscript, the authors should extensively review the existing literature on the movement of powdered substances or particles on water surfaces through irradiation as well as other forms of nano-locomotion. This comprehensive review should include, but is not limited to, studies such as Norikane et al. *Cryst. Eng. Commun.* 2016, 18, 7225; Fujii et al. *Adv. Funct. Mater.* 2016, 26, 3199; and Lucchetta et al. *AIP Adv.* 2015, 5, 077147.

We would like to thank the reviewer for pointing out that we may address this area of research more extensively in the manuscript. However, we would also like to point out that the phenomenon reported here should not be considered a form of nano-locomotion. This becomes apparent from the facts that:

- 1) individual, sparsely distributed particles do not undergo any directional motion (result sections 1-2) even if irradiated by a similar Gaussian profile as in the flow movies of e.g., Supplementary Movie M9.
- 2) In absence of light the particles in the flow region are observed to be rigidly assembled by capillary forces, an aggregated state without particle diffusion in which they would be unlikely to undergo nano-locomotion under illumination (see detailed description in answer to question 3.7).
- 3) The collective deformation of particles can lead to objects orders of magnitude larger than the particles being pushed along the flow trajectory. We occasionally observed this when pieces of dirt were present at the liquid interface, as shown in the Figure below. Rather than seeing the large object as an obstacle, the particle assembly pushes the object through the flow field *at the same speed* as the displacement of surrounding particles, indicating a collective deformation process.

Figure. Dragging of larger passive objects. Snapshots of a y -oriented flow induced by a laser beam with y -oriented polarization, similar to the results shown in Figure 4c without dirt piece, at different timepoints. Here, frames were chosen surrounding the event of a large piece of dirt entering the field of view and being dragged along the flow (observed for this size of passive object only once). The different frames show the total flow just before the entrance of the dirt piece (a), when the dirt piece enters along the inflow direction from the left (b), when it moves across the top-left quartile following the flow arrow trajectory (c-f), and just before (g) and after (h) exiting the field of view via the top. Note that the passive object's velocity corresponds to the particle velocities. Rather than seeing the large object as an obstacle for a free-particle flow (such as in individual particle nanolocomotion), the assembled particles seem to collectively push the dirt piece across the field of view at a velocity thus corresponding to their own.

In view of this nuance and inspired by the helpful questions raised below by the referee, we decided to include references to some nano-locomotion effects in an additional paragraph in section 3. In this paragraph, we then discuss why the observed phenomenon cannot be considered a nano-locomotion effect but rather is an effect due to the photodeformation of *assembled* particles, i.e., a synergy between local, illumination-triggered particle assembly and polarization-dependent particle photo-deformation itself.

Finally, we think novelty is ensured by the fact that none of the previous approaches induce symmetry-breaking by means of the illumination's polarization. For example, the articles mentioned by the reviewer obtain directionality from tilted incidence with respect to the interface or from intensity gradients associated to rather focused laser spots. Whilst we also have intensity gradients, these do not directly relate to the flow pattern and the latter can be altered without changing the intensity pattern (see detailed description in answer to question 3.3). Furthermore, we note that the speeds reported here (up to 90 $\mu\text{m/s}$ for mostly sub-micrometric particles, see Supplementary Movie M16, Supplementary Figure S16 and section 4), which translate into ~ 100 body sizes per second, imply a mechanistic difference with respect to the ~ 10 body sizes per second reported in the suggested articles. Polarization as a symmetry-breaking mechanism contrasting with other approaches is discussed in a wider context, involving particle displacements in various media, in the Discussion section.

Text modifications: We clarify more clearly in section 3 that the effect is not one of nano-locomotion, amongst others by mentioning different kinds of such effects as suggested by the referee. We are open to including the example of a larger object being pushed by the assembled particles in the corresponding outlook at the end of the article (perspective of controlling displacement and orientation of passive objects), to give the reader an additional cue about the nature of the effect at display, if deemed helpful from a technical point of view.

2. What method did the authors employ to achieve stable dispersion of individual azobenzene polymer particles at the water-air interface? Typically, particles tend to aggregate on water surfaces or disperse in the water. Successful stable dispersion of azobenzene polymer particles is a key technique in this research. Consequently, the authors should provide a detailed explanation of their technique for creating a stable dispersion on the water surface, supported by experimental evidence, such as water contact angle measurements.

We thank the reviewer for highlighting the importance of achieving a stable dispersion of azobenzene polymer particles at the water-air interface. Indeed, preventing aggregation or desorption into the bulk is essential for the reproducibility of the presented experiments.

To address this challenge, we employed the method described in our recent publication (Rey et al., Versatile strategy for homogeneous drying patterns of dispersed particles, *Nat Commun* 13, 2840 (2022)). This approach enables the preparation of interfacially active and stably dispersed particles by tuning the surface chemistry during synthesis. In particular, polyvinylpyrrolidone (PVP) or polyvinyl alcohol (PVA) is used as a steric stabilizer, which enhances colloidal stability in water and promotes particle adsorption at the air-water interface. Once adsorbed, we did not observe any desorption over time, consistent with earlier findings on irreversible interface adsorption (e.g., Pieranski, P. *Physical review letters*, 45(7), 569 (1980)).

The stabilized particles remain well-dispersed at the interface and undergo lateral Brownian motion without noticeable aggregation. This behavior is consistent with previous observations of PVA-coated polystyrene particles (see Rey et al., 2022, specifically Supplementary Movie 11). The stabilizing effect appears sufficient to counteract attractive capillary interactions. Although the exact physical mechanism underlying this unexpectedly strong interfacial stabilization by steric layers is not yet fully understood, we are actively investigating this aspect and hope to clarify it in future work.

We also note that in early trials, we tested sodium dodecyl sulfate (SDS) as a surfactant for interfacial stabilization. However, as the reviewer suggested, this approach resulted in strong aggregation at the interface and was unsuitable for our study. We therefore focused exclusively on sterically stabilized systems.

To support the interpretation of interfacial trapping, we measured the three-phase contact angle of the particles using the gel trapping technique developed by Paunov (Paunov, V. N. (2003). *Langmuir*, 19, 7970–7976). We obtained a contact angle of $111^\circ \pm 4^\circ$ or $104^\circ \pm 6^\circ$ for the PVA and PVP stabilized polymer particles, resp., which is consistent with previously reported values for polymeric particles at the air-water interface (see e.g., Maestro,

A. et al., *Current opinion in colloid & interface science*, 19(4), 355-367 (2014) and Arnaudov, L. N. et al., *Physical Chemistry Chemical Physics*, 12(2), 328-331 (2010).

Text modifications: We added the Figure below as new Supplementary Fig. S2 (also with respect to Referee 1's question 3). We briefly mention this aspect of our method in the main text and reference the previous work. We mention that a stable dispersion is necessary to obtain the flow effect in section 3, as the flows require particle mobility for non-illuminated areas during capillary assembly deformation.

New Supplementary Figure S2. Three-phase contact angle by gel trapping. a) Schematic illustration of the determination of the three-phase contact angle by the gel trapping technique developed by Paunov (Paunov, V. N. (2003). *Langmuir*, 19, 7970–7976) with an atomic force microscope (AFM; for details see Methods). AFM height image with cross-section at the indicated white line for (b) Sample 1, and (c) Sample 2. d) Box plots of the determined contact angles for either sample type.

3. The authors should discuss the photoisomerization of the azobenzene polymer particles more carefully.

3.1. The authors should clarify how the laser wavelength correlates with the absorption of the azobenzene polymer. They must also include the time evolution in the UV-vis absorption spectra of the azobenzene polymer before, during, and after laser exposure. This information is crucial for analyzing the light-induced properties of azobenzene-based materials and represents one of the most fundamental datasets in this study.

We realize that we may not have included information about the azobenzene-related processes sufficient for readers with an interest in the molecular processes. As mentioned above, Disperse Red 1 (DR1), is a pseudo-stilbene, with strongly overlapping *trans* and *cis* spectra (Mahimwalla, Z. et al., *Polym. Bull.* 69, 967–1006 (2012)). The wavelength chosen (532 nm) is a rather standard one to induce continuous photo-cycling between the two

isomers, in a spectral region where the absorption is similar for both isomers. The differential absorption spectra of both isomers of DR1 can for example be found in Poprawa-Smoluch, M. et al., *J. Phys. Chem. A* 110, 11926–11937 (2006). Note that this contrasts with a large body of work using other azobenzene classes for targeted *trans* to *cis* or *cis* to *trans* switching, often by employing two separate wavelengths. When using polymers incorporating pseudo-stilbene type molecules for isomer cycling at a single wavelength, the suitable wavelengths are in the blue/green region (Natansohn, A. & Rochon, P., *Chem. Rev.* 102, 4139–4176 (2002)), and most works we are aware of use a single wavelength between 491 nm and 532 nm. Due to preferential absorption of the *trans* molecules for electric field vectors parallel to their axis, prolonged cycling can lead to a statistical reorientation effect driving a polarization-directed deformation. These effects are nicely summarized in (Priimagi, A. & Shevchenko, A. *Journal of Polymer Science Part B: Polymer Physics* 52, 163–182 (2014), accessible summary, review focusing more on microfabrication applications and variations of the concept) and in (Saphiannikova, M. et al., *Soft Matter* 20, 2688–2710 (2024), ref 25 in the manuscript, more recent and focusing on the theoretical aspects mainly).

The reviewer is right that isomerization-related changes are expected under illumination. As for other azobenzene dyes, the thermal equilibrium population consists of mostly *trans* isomers. Therefore, a non-negligible *cis* population is created as a side-effect of *trans-cis-trans* isomer cycling. The lifetime of the *cis* state for pseudostilbenes such as DR1 is usually on the order of seconds (considerably shorter than azobenzene-type and aminoazobenzene-type molecules, see e.g., Mahimwalla, Z. et al., *Polym. Bull.* 69, 967–1006 (2012)). Importantly, no important differences in absorption and thus isomer populations or photo-thermal effects are known for different polarizations.

To briefly illustrate the (known) absorption signature of the DR1 *cis* population arising during and decaying after exposure in our setting, we dried particles on a glass slide and monitored the transmission changes induced by both linearly and circularly polarized 520 nm laser light at two different reading wavelengths (570 nm and 600 nm). From the measurements in the Figure below, the following observations can be made:

- 1) The sign of the transmission changes during illumination is consistent with the differential transmission spectra of pseudostilbenes, which usually have a higher *cis* absorption in the long wavelength region (around 600nm), as opposed to a much higher *trans* absorption coefficient at for the main peak around 500nm (see Poprawa-Smoluch, M. et al., *J. Phys. Chem. A* 110, 11926–11937 (2006) for DR1).
- 2) The *cis* population's transmittance signature takes about 10 seconds to fully decay, consistent with measurements in literature on pDR1m-co-mma polymers (Barrett, C. et al., *Macromolecules* 27, 4781–4786 (1994)). Note that regarding the flows, this is too long to explain the abrupt flow stops when illumination is shut (if the flows were driven by *cis-trans* gradients) and regarding the capillary assembly part, too short to explain continued particle bonding present even minutes after illumination is turned off.
- 3) No obvious difference is found in the effect induced by linearly and circularly polarized laser illumination. We therefore postulate with regards to the following questions that any effects rooted in photo-thermal behavior or isomerization itself should be independent from the use of linearly or circularly polarized light, all other experimental conditions maintained.

Text modifications: We explain the known underlying mechanisms of photo-deformation in the polymer employed more thoroughly in manuscript section 1. Although they are not directly relevant for the higher-level effects reported here, we have become aware that confusion should be avoided specifically with respect to parts of the nano-locomotion field and others which employ azobenzene-type molecules (such as e.g., some azobenzene surfactants). We also point to the thermal *cis* lifetime in section 3.

Figure. Relative transmission changes during and after exposure to laser light. a/b) Transmission change evaluated for two wavelength regions during and after a short exposure to linearly/and circularly polarized light. The absorption linked to *cis* isomers is higher around 600 nm and above, and lower at shorter wavelengths, consistent with literature. The decay takes about 10s in total. No major difference is observed for the two polarizations, as expected. Note that dedicated studies to this effect, for this polymer and similar ones, report that the decay can have several components (Barrett, C. et al., *Macromolecules* 27, 4781–4786 (1994) for this polymer type and Barrett, C. et al., *Chem. Mater.* 7, 899–903 (1995), for similar DR1-containing ones), but overall report similar decay timescales and similarly looking results.

3.2. More quantitative analyses of the particle morphology changes and movements are necessary. The wavelength and intensity of the light source typically influence the light-activation properties of azobenzene-containing materials.

We agree that a more quantitative evaluation of the deformation effects shown in Figure 1 may be helpful, especially since the shape is relevant to the capillary processes. To this aim, we acquired 60 movies showing the deformation of an individual particle, large enough to be quantitatively analyzed, being deformed (20 cases for each illumination condition, i.e., linear, elliptical and circular polarization at fixed intensity). We then detected the particle outlines for each frame using the segmentation procedure described in the Methods section *Particle detection at interfaces*. We then extracted the values both for the particle area and for its aspect ratio, as a function of (illumination) time. The results are shown in the New Supplementary Figure S3 below. Generally speaking, the trend of higher to lower aspect ratio from linear to circular polarization shown in Figure 1 is confirmed. One can also observe that all photo-deformations lead to in-plane expansion, this effect being strongest for circular polarization. Finally, the deformation process is continuous, making illumination time relevant for the total extent of deformation.

Regarding the intensity and wavelength, the following should be noted: 1) Although the wavelength to produce photo-deformation can be varied for these azopolymers, this does not produce fundamentally different *trans-cis-trans* cycling effects. For example, if the wavelength was shifted closer to the absorption peak of the *trans* molecule the cycling would become less efficient compared to the absorption and the absorption length would decrease (Januariyasa, I. K. et al., *ACS Appl. Mater. Interfaces* 15, 43183–43192 (2023).), an effect not necessarily beneficial for this work. Both *trans* and *cis* spectra for DR1 can be found for example in Dumont M. L. et al., *Proc. SPIE 2042, Photopolymers and Applications in Holography, Optical Data Storage, Optical Sensors, and Interconnects*, (1994); <https://doi.org/10.1117/12.166297>), and the *trans* absorption spectrum of our exact polymer in Urban, D. et al., *Nat Commun* 14, 6843 (2023), in Supplementary Figure S1b of the article. Intensity on the other hand is usually known to linearly increase the deformation speed (as the cycling process gets accelerated linearly with more light, see e.g., Ryabchun, A. et al., *Advanced Optical Materials* 7, 1901486 (2019).). In our studies of the deformation effect at liquid interfaces we did not notice any special behaviors with respect to the intensity employed, however we placed ourselves in a range where particle deformation occurs fast enough

to not be dominated by rotational particle diffusion and slow enough to not induce particle disintegration. In the movies with stripe-shaped illumination, at 645 W/cm^2 , we get close to this condition, as can be for example seen in Supplementary Movie M14 (left), for some of the particles in the stripe region.

New Supplementary Figure S3: a) Mean aspect ratio vs. time of particles at the interface under illumination with linearly, elliptically and circularly polarized light. As exemplified in Figure 1, the induced in plane aspect ratio is highest for linear polarization, while for circular polarization the value remains close to one. b) Similar graph, reporting the mean relative area expansion vs. time. Here, circular polarization is found to induce the largest net flattening, albeit considerable area expansion is also produced in the other cases. In both graphs, error bars report the standard error of the mean (s.e.m.) for 20 individual particles for each polarization, while the color-shaded regions represent the distribution width based on the corresponding standard deviation (s.d.). Note that while a clear statistical difference can be observed between the different sample populations, there is also a considerable distribution overlap, especially between linear and elliptical polarization. This can supposedly be explained by the polydisperse sizes of the particles, their random rotational diffusion under illumination as well as fitting noise. c) Example images at different time points of the fitting procedure for a movie showing a particle deformed with linear polarization. Top-to-bottom: movie frame, corresponding segmented frame, and movie frame with fitted ellipse overlay. The fitting procedure is detailed in the method section. All stacks and corresponding segmented versions with values are included in the shared data folder. Intensity $I = 219 \text{ W/cm}^2$.

Text modifications: The results for the individual particle deformation were included as New Supplementary Fig. S3 and discussed at the end of section 1. We thank the reviewer for pointing this issue out, as we think this aspect contributes nicely to the intelligibility of the subsequent result sections. We also explicitly indicate the useful intensity range in the Methods section, to facilitate ease of reproducibility.

3.3. The factors affecting capillary forces should be discussed more carefully. The surface free energy, as well as the shape of the particles, influence capillary forces. During $n-\pi^*$ absorption, azobenzene is known to undergo trans-cis conformational interconversion. This photoisomerization process results in two notable effects, in addition to shape alterations in azobenzene materials. First, the photothermal effect occurs, where irradiation elevates the temperature of azobenzene materials. Second, there is an increase in the composition of cis-isomers, which exhibit greater hydrophilicity than trans-isomers. Both temperature changes and increased hydrophilicity can alter the surface free energy of azobenzene polymer particles. These phenomena should be addressed in the Results and Discussion sections. For instance, when a linearly polarized 532 nm laser irradiates one side of the azobenzene polymer particles, it can create a gradual difference in the surface free energy across the particle surface, resulting in nano-locomotion on water surfaces.

We acknowledge that the reviewer raises very relevant questions here and realize that we have not formulated our reasoning with respect to these questions clearly enough in the initial draft. The reviewer is correct, that both photo-thermal effects and a higher *cis* population are likely to occur at the intensities used. In addition, we have a Gaussian laser beam profile in all experiments except for those related to Figure 4e-g, thus symmetric intensity gradients are naturally present. Given that we do not see persistent locomotion in the case of sparse particles, we believe that this may be attributed to particles ($<3 \mu\text{m}$) being rather small compared to the length scale of the gradient (Gaussian beam diameter $\sim 70 \mu\text{m}$ in all sparse experiments). In the articles about manipulating particles directly with intensity gradients, it seems that laser spots are of more comparable size to the objects moved. Potentially, the steric stabilizer shields the azopolymer particles hydrophilicity changes to a certain extent.

What we can say with certainty is that such effects can be excluded as causes of the flow phenomenon when considering the polarization-dependence of the flows. Note that the intensity pattern of the spot in Figure 3 and Figure 4a-d is axially symmetric (Gaussian beam), which is not reflected by the symmetry-broken directional flow patterns. In addition, Figures 4c and 4d show flows obtained from the same – still axially symmetric (see inset Figure 4a) – intensity profile where the symmetry breaking is altered by changing only the polarization direction. As shown in Supplementary Figure S11 and Supplementary Movie M11, we can even alter the flow direction during the flow by simply rotating a waveplate (and thus the polarization), without modification of any other parts of the optical set-up. Thus, not only does the flow pattern break the symmetry of the intensity pattern, but the flow directionality of the pattern can be altered *without* modifying the intensity pattern. Finally, the stripe intensity pattern shown in inset of Figure 4e leads to pure shear patterns when the polarization is oriented diagonally to the stripe (see Supplementary Movies M15 and M16 and Figure 4 e-h). In our opinion, there is no compelling link whatsoever between the flow pattern and the intensity pattern here, as one should expect for photo-thermal or isomerization driven flows. To visualize this aspect better at first glance, we prepared the new Supplementary Figure S15 below, which tracks the movement of two prominent particles from Supplementary Movie M15, as they move in opposite directions across the whole field of view alongside the symmetric, stripe-shaped illumination pattern.

New Supplementary Figure S15. Particles moving in opposite directions under shear flow. Frame showing the field of view during the shear flow induced by diagonal polarization in Supplementary Movie M15 (right), at raw data time $t = 70\text{s}$, corresponding to 17.5 s in the accelerated movie. The trajectory of the two encircled prominent particles was tracked as they crossed the field of view in opposite directions (see Methods for the tracking procedure). The colorbar to the right indicates the time difference between the particles' passage at a position labeled with a correspondingly colored circle, and the current frame time. Red arrows schematically depict the flow direction. Green overlay: schematic illustration of the stripe-shaped illumination. Scale bar (white): $20\ \mu\text{m}$.

Finally, to further corroborate the argument above, we provide additional data in flow condition (i.e., at sufficient particle density) showing the difference between illumination with linear, elliptical and circular polarization (included as new Supplementary Movie 12 and new Supplementary Figure S12, which is reproduced below), while all other parameters are maintained. Note that, as detailed in response to question 3.1, no significant difference is expected with respect to photo-thermal heating or isomerization dynamics, just because the polarization is altered between linear to circular. The absorption profile, related heating, and isomerization effects are the same. What changes, e.g., for circular polarization, is the deformation of the particles (in-plane flattening instead of directional deformation, and thus no capillary interaction, see also response to question 4 from Reviewer 3). We observe that for circular polarization, no continuous flow develops at all, despite the intensity pattern, also here, being unaltered with respect to the linearly polarized and the elliptically polarized examples.

New Supplementary Figure S12. Flow attempts with non-linearly polarized illumination. Flow fields from the illumination events shown in new Supplementary Movie M12, using linearly (left), elliptically (middle) and circularly polarized (right) illumination. Flow fields are shown before illumination (a) and averaged over adjacent 10 second intervals after illumination is turned on (b-d). While a tiny transient response is visible for all samples in b, no sustained flows develop for the circularly polarized illumination d (right), whilst linear polarization leads to the strongest sustained flow d (left). e) Zoomed snapshots from the movie at different timepoints showing a prominent particle before illumination and after illumination was turned on. The snapshots allow to recognize the characteristic deformation of the particles, particularly the disk-like shape under circularly polarized illumination. Intensity $I = 56 \text{ W/cm}^2$.

Text modifications: The above data for illumination in flow conditions with different polarizations (linear, elliptical, circular) is provided as additional Supplementary material and referenced in section 4. The visualization of the tracked particle positions in shear flow condition is included as Supplementary Fig. S15, with a cropped version being inserted as inset into Figure 4f. Further, we tried to reformulate certain aspects of the polarization-dependence of flows in this section and section 3, to make the arguments more compelling.

3.4. To substantiate the claim of "shape-mediated assembly and disassembly" as discussed in the Discussion section, where the authors claimed that capillary forces resulting from rod shapes would be the sole cause of continuous locomotion, they must thoroughly rule out alternative explanations. This includes eliminating the possibility of other effects, such as Marangoni propulsion, which has been extensively researched in the field of nano-locomotion. The authors should address this issue using appropriate scientific methodologies.

Here we need to stress that in our picture of the effect, we do not attempt to claim that capillary forces resulting from the particle shapes are the sole cause of continuous locomotion. We believe that the rod-like particle shapes lead to assembly, as shown in the sections preceding the flows. Once assembled into aggregates covering the whole illumination zone, the particles propagate their deformation across the entire assembly (in other words, the whole assembly deforms). During this process the particle shapes change, the assembly remodels, and new particles freely diffusing particles are entering the assembly. Thus, the continuously growing, deforming particle assembly manifests itself as a continuous flow phenomenon.

In principle, the polarization-dependence arguments provided in response to the previous question, also cover Marangoni and other effects at a global level. Neither the difference between linear and circular polarization, nor between different linear polarizations, nor the shear flow patterns from stripe-shaped illumination can be conveniently explained by photo-thermal or isomerization-based mechanisms, whose flow patterns should be in related to the effectively delivered light intensity pattern (only). In particular, maintaining the same symmetric Gaussian illumination profile, the flow direction can be altered by changing only the linear polarization direction and no flows develop if the polarization is circular. In addition, as pointed out by the reviewer below (question 3.8), the particles in absence of light are firmly assembled and not individually diffusing, which would seem like a difficult condition to start smooth Marangoni flows from.

However, we would also like to point out that when the illumination first hits the particles at the beginning of a flow movie, when they are still freely diffusing on the interface and before their capillary assembly, a weak, transient, in-plane isotropic photothermal flow can be observed. This observation is reported and explained in more detail in the answer to the partially overlapping question 2 by Reviewer 3.

Text modifications: We tried to streamline our explanation of the flow phenomenon in the manuscript (section 3), in particular putting more emphasis on the assembled state of the particles, as well as discussing the matter of polarization-dependence (and thus exclusion of photo-thermal or isomerization-related flow phenomena) in section 3 and 4.

3.5. The authors should distinctly differentiate between nano-locomotion, which results from microscopic forces near particle surfaces, and macroscopic water movement. The initial part of the Results section noted that the azobenzene polymer particles aggregated together after changing from spherical to rod-shaped, even after the laser was turned off. This indicates that the capillary forces from the rod shapes promote aggregation rather than repulsion. Figure 3iii and Movie M7 (note: Movie M9 in revised version) provide clear evidence of rod-shaped azobenzene polymer particle aggregation even during irradiation. Concurrently, the clustered particles were drawn away from the irradiated area on the water surface. The authors suggested that water flow caused by osmotic pressure was a potential cause, inappropriately citing photoswitching of the adsorption/desorption behavior of azobenzene surfactants. They claimed that photoisomerization of azobenzene components increased hydrophilicity, creating the "sustained pumping machinery." However, this effect is highly unlikely in the current system, where no azobenzene derivatives are dissolved in the aqueous phase or detached from the particles; therefore, a local concentration of cis-isomers and the resulting osmotic pressure should not be generated. Consequently, this study lacks a compelling explanation for the continuous flow observed during irradiation.

We acknowledge that the paragraph in question may have been misleading and sincerely apologize for the confusion. We have removed the paragraph in question.

The initial aim of the paragraph was by no means to suggest an equivalence of mechanisms (i.e., osmotic pressure driving the flow). As the reviewer rightly points out, this would be highly unlikely, since the azobenzene-bearing moieties are covalently bound as side chains to a hydrophobic linear polymer (MW ~ 10 kDa for this azopolymer as measured by us in a previous publication, see Urban, D. et al., *Nat Commun* 14, 6843 (2023)).

Text modifications: This paragraph is removed entirely from section 3 and replaced in length by a paragraph treating the exclusion of other, known flow phenomena, as outlined in the answer to the two preceding questions.

3.6. As illustrated in Figure 3 and Movie M7 (note: Movie M9 in revised version), the nano-locomotion effect was the most pronounced in the central region of the scope. This observation suggests that the intensity of the laser progressively decreased from the center to the periphery of the exposed area. Such a gradient in laser power could also explain the observed continuous flow.

We thank the reviewer for their thoughtful comment. The laser illumination in Figure 3 and Supplementary Movie M9 (updated numbering) indeed has a Gaussian profile, is centered in the field of view, and covers about half its area (spot diameter ~80 μm). However, we do not observe the strongest particle motion at the center of the beam. As shown in Figure 4c,d (same experiment as Figure 3 but with smaller particles), the displacement velocity increases across the laser spot and reaches its maximum near the edge of the field of view.

This suggests that the motion is not driven by local gradients of light intensity. Instead, it supports the interpretation that the flow results from capillarity-driven assembly and cooperative photo-deformation propagating across the entire assembled zone. As the deformation of individual particles gets propagated, the contribution of the local intensity driving local photo-deformation gets integrated from the center of the periphery towards the border of the spot. Note that at the boarder of the field of view, the displacement velocities are still equal to the velocities at the boarder of the illumination spot.

We believe that also the flows shown in Supplementary Fig. S14 can be of some interest here. In fact, for an x-oriented stripe illumination, we observe significantly higher outflow velocities for x-oriented polarization (polarization parallel to the long stripe axis) than for y-oriented polarization (polarization perpendicular to the long stripe axis). In principle, the gradient perpendicular to the long stripe axis should be steeper. However, in the deformation propagation picture, the outflow velocities should be higher if deformations can propagate over longer distances, i.e., along the illumination stripe, which is the case observed.

Finally, regarding the role of intensity patterns more generally, we refer to the discussion in our response to Question 3.3, where we explain why neither photothermal nor isomerization-based mechanisms can account for the observed flow directionality and its polarization dependence.

Text modifications: Same as question 3.3. In addition, we added information about the Gaussian beam profile in the text of section 3, which was previously only schematically illustrated in Figure 3.

3.7. Additionally, when the light-induced motion of azobenzene materials typically ceases upon turning off the light source, the photothermal effect of azobenzene groups during illumination is generally considered the primary cause of this movement. Although hydrophilic *cis*-isomers are transformed back into hydrophobic *trans*-isomers under dark conditions, this conversion does not occur instantaneously after the light source is deactivated. As previously mentioned, the time-dependent changes in UV-vis absorption spectra would likely support this gradual conformational reversal process.

We like the idea of considering time scales in the discussion about which effects drive the flow phenomenon. As the reviewer correctly points out, the time scale of *cis* to *trans* relaxation (here ~ 10 seconds) is too slow to account for the immediate flow interruption when the laser light is blocked. If a local, more hydrophilic *cis* population (for example on one side of each particle) was driving the flow, this population would not suddenly disappear upon laser blocking and hence the flow would continue for a timespan comparable to *cis* \rightarrow *trans* relaxation, an argument also used in the corresponding article (Norikane, Y. et al., *CrystEngComm* 18, 7225–7228 (2016) suggested by the reviewer in the above.

On the other hand, the reviewer will certainly acknowledge the fact that the flow rise time scales are not consistent with a photo-thermally driven phenomenon either. In fact, as shown in Figure 3d, the first time the illumination spot is applied to pristine, freely diffusing particles at the interface, the flow needs 20 seconds to rise to its constant value. On the other hand, when illumination is interrupted for ~ 20 seconds, it jumps to its maximal value immediately after illumination is resumed. We would stipulate that for an absorption-driven photo-thermal effect the flow dynamics should be independent of prior illumination. In section 3, we explain how this is linked to our explanation of the flow mechanism, namely the pushing of locally assembled particles onto each other, while they are continuously being photo-deformed. In particular, the first time the laser spot hits the pristine particles, they deform, then assemble into a local particle aggregate, and only then are able to propagate their respective deformation over a larger length scale (across the laser spot). When illumination hits a pre-assembled (previously illuminated structure), the particles are already assembled and hence can start propagating the photo-deformation instantly.

Text modifications: We included and refined the above time-scale arguments in our description of the flow effect in section 3.

3.8. The rod-shaped particles aggregated after changing their shape, regardless of the irradiation. This suggests that deformed particles expelled from the irradiated area would aggregate together once the laser was turned off. However, Movie 7 (note: Movie M9 in revised version) demonstrates that the deformed particles remained stationary on the surface of the water after laser deactivation. This raises the question of what accounts for this discrepancy.

As the reviewer correctly points out, transformation into a rod-like shape is expected to lead to capillary assembly of particles. This is in fact what is observed during the early illumination stage in Supplementary Movie M9 (and shown to the best of our capabilities in Figure 3c). The particles do not undergo visible assembly effects after the illumination is turned off since they already are assembled. We believe that the apparent discrepancy comes from the fact that the smaller particles of the distribution (see Supplementary Fig. S1a,c,e) are below the optical resolution limit and that particle deformation with associated flattening (quantified in the answer to question

3.2) leads to a reduction in contrast, so that the continuous nature of the assembled particle structure is difficult to appreciate by eye. The reasons for contrast loss upon flattening were included in the response to reviewer 1's question 3. The major reason for the authors to include the data shown in section 3 (in addition to the data in section 4, obtained from smaller particles giving rise to smoother and more homogeneous flows) was in fact that in this case, the capillary assembly process involved in the flows can at least partially be appreciated directly by eye.

The fact that particles in the illumination zone (as well as above and below the latter for vertical polarization after sufficient flow time) are assembled can be appreciated in the illumination-off situation via the fact that individual particles do not undergo individual diffusion anymore, but only faint diffusion in large patches (Supplementary Movie M9). When illumination is on, this observation is more difficult as the particles are continuously deformed and new particles are recruited into the assembly, which thus continuously remodels. Also, characteristic structures of capillary assembly (side-by-side stacking) are still visible in the quasi-static aggregates. It should be highlighted that after sufficient flow time, the aggregate of assembled particles extends beyond the field of view (for example in the vertical direction for vertical flows), since there is no passive mechanism for disassembly of deformed particles when not illuminated, i.e., when exiting the illumination zone. We can illustrate this by additionally providing images that were acquired while looking at the areas above and below the field of view after the flow sequence shown in Supplementary Movie M9 (see figure below). These images show the vertical extent of the capillary assembly, extending into areas that were never illuminated during the flow event. We include this set of images as new Supplementary Fig. S10 and added the movie of the experimenter moving the field of view over the total aggregate to the available shared data set.

New Supplementary Figure S10: a) Scheme of how the capillary assembly of particles extends beyond the field of view (here in y -direction for y -polarized illumination) after prolonged capillary assembly and deformation-driven flow. b-e) Roughly adjacent frames showing the areas above and below the illuminated zone in Figure 3/Supplementary Movie M9. Circles with same color label the same features to provide visual orientation. The illumination zone during the flow was smaller than the field of view and approximately corresponding to the lower half of (c) and the upper half of (d). The underlying movie of the experimenter moving the field of view across the particle assembly, in which still no individual particle diffusion is observed, is included in the raw data as:

«001_M9_LargeParticles_YPOL_50fps_EXTRA_OV_afterFlow.avi»

Note also that particles that were previously assembled and expelled from the illumination zone can occasionally be seen to re-enter the illumination zone (still assembled) depending on the polarization sequence. For example, when attentively studying Supplementary Movie M11 (rotating the flow axis by step-wise rotation of the polarization), one can recognize a thin strip of previously assembled and expelled particles (non-diffusing) re-entering the field of view from the top/top-left and contributing to the flow a second time, from 00:23 on (please note that this nuance is more easily appreciated in the corresponding raw data movie “006_M9_Rotpol_50fps.avi” (01:15-01:45), as the movies had to be reduced in size for submission, raw data snapshots are provided in a second figure below for illustration of the observation).

Figure. Previously expelled particle assembly parts re-entering the field of view. Sequence of raw data snapshots (top) with corresponding frame of the flow field movie (Supplementary Movie M11) during a flow with stepwise rotation of the polarization. At around 01:20 the polarization is shifted from horizontal to diagonal, changing the instant deformation direction of the continuously assembling aggregate (blue ellipse). At about the same time, a previously assembled part of the aggregate, with already deformed and non-diffusing particles can be seen to re-enter the field of view from the top left (orange ellipse). Eventually the two capillary assembly parts merge and the flow resumes as usual. On the bottom, a series of schemes illustrates how the arms of the continuously growing aggregate may get sucked in again by the inflow orthogonal to the deformation when the polarization is continuously rotated during flowing, basically tying a knot. Note that the actual situation appears slightly less symmetric than in the proposed scheme and assembled particles from the bottom-right arm can only be seen to enter the field of view (from the right) around 01:55 in the shared raw data movie.

Text modifications: The figure showing the extents of the assembled aggregate pertaining to the flow event in Figure 3 and Supplementary Movie M9 is added to the Supplementary Information and referenced alongside the

discussion of the capillary assembly and flow in section 3. The corresponding movie showing the aggregate extents is added to the shared data folder. The aspect of particles being assembled in the illumination zone (due to shape-mediated capillary forces) and staying assembled even when leaving the illumination zone is highlighted more clearly in the text.

Reviewer #3 (Remarks to the Author):

The manuscript entitled “Directional capillary flows induced by photo-deformable colloidal particles” by Urban et al. presented very interesting assembly and disassembly of azopolymer-based shape-deformable microparticles at the air/water interface in response to polarized light. However, to precisely evaluate this work, the capillary flow-driven mechanism should be clarified.

We appreciate the reviewer’s work in formulating precise and knowledgeable questions, not only about the flow phenomenon but also concerning the assembly process. As some of the questions overlap heavily with those raised by other reviewers, we occasionally refer to the previously provided answers for the sake of conciseness.

(1) The authors applied highly intensive polarized light to drive the shape deformation. In addition to photoisomerization, azobenzene-based aggregates would perform photothermal conversion under light irradiation, which enables driving directional movement at the air/water interface as well (e.g., CrystEngComm, 2016, 18, 7225). The authors should clarify whether there is a photothermal behavior or not. And the impact of this behavior on the directional assembly and particle flow.

1) Regarding impact on flows (section 3 and 4)

As outlined in our answer to reviewer 2’s question 3.3, we observe no persistent directional flows for sparsely distributed particles as in section 2, although the laser profile is a Gaussian beam of similar extent (~70 μm) also there. Therefore, flows arising from thermal gradients across the particles themselves do not seem to drive the phenomenon (for a more global photothermal effect, see answer to the next question).

2) Regarding impact on assembly and disassembly

Regarding the assembly and disassembly process, it should be highlighted that linear and circular polarization lead to similar absorption and hence to a similar photothermal and isomerization contributions (see answer to question 3.1 by reviewer 2). It would therefore be unlikely to see fundamentally different behavior for the two different polarizations, since from an absorption and isomerization point of view there is only “illumination on” or “illumination off”. We also point out that for Supplementary Movies M2-M8 (assembly and disassembly sequences), the wide Gaussian laser beam is not necessarily exactly centered on the particles shown. We have not observed any effects related to the position of the particles with respect to the point of highest intensity of the laser beam, as would be expected for gradient-driven phenomena.

Text modifications: The absence of persistent flows for sparsely distributed particles is highlighted in the text of section 3, when discussing the appearance of the collective flows. The similar effects from a photo-thermal and *cis* isomer population point of view for differently polarized light is mentioned at the end of section 1.

(2) Similarly, such highly intensive light is another possible source producing a thermal effect on the water, which may drive particle flow. The authors should clarify its impact on the water and strong photoabsorbing azopolymer particles.

The reviewer raises a very valid point here regarding the heating of the total Gaussian laser spot area and its potential consequences. We note that even if water absorption at visible green wavelengths (532 nm) can be considered negligible, the strong photo-absorption of dense particles could indirectly map the intensity profile into a temperature profile on the water surface, driving e.g., a global Marangoni flow. This could then also explain why densely packed particle distributions would undergo thermal flow motion, while individual particles would not. Namely, in this case the flow displacing an individual particle would be driven not by the heat generated by itself, but by the heat generated by all the other surrounding particles that absorb illumination.

As a matter of fact, we transiently observe such a small photothermal effect at the very initial stages of the illumination onset, while the particles are still freely diffusing at the interface, for example in Supplementary Movie M10. The tiny, in-plane isotropic flow contribution can be appreciated at 00:02-00:03 in the Supplementary movie as small outward pointing arrows. In the figure below, we plotted the corresponding flow fields rescaling the arrows to highlight this small contribution better (even if still, the effect is at the edge of what we can be detected with the home-made detection algorithm that we wrote specifically for this work). As soon as particles are assembled, the polarization-driven anisotropic flow phenomenon takes over. Note that we also see this transient isotropic Marangoni contribution as a spike in relative flow divergence (see graphs in the provided figure below).

Figure. Transient photo-thermal component. a,b) Sequential flow fields of the x- and y-directed flows induced by linearly polarized light shown in Supplementary Movie M10. The frames show the detected flow pattern before illumination (left) and after illumination onset (second from left to right) averaged over consecutive 3.3 s intervals, for x-oriented polarization in (a) and y-oriented polarization in (b) respectively. On the second frame from the left, a barely detectable transient radial velocity profile is observed in both cases, which appears to be a polarization independent displacement induced by the laser onset. On subsequent frames, the persistent, polarization-directed flow behavior emerges. Note that the velocity arrows were scaled up with respect to the arrows shown in the movie and in Fig. 4 c,d, to highlight the small, transient contribution. Here, the reference arrow (gray) stands for $5 \mu\text{m/s}$, rather than $20 \mu\text{m/s}$ in Figure 4 c,d. Scale bars (black): $20 \mu\text{m}$. c) Graphs plotting the relative flow divergence as a function of time for both cases, showing a spike for the timestep with the transient contribution. Indeed, such a radial profile is highly divergent, as there is only outflow.

On the other hand, the polarization-dependence of the flows developing afterwards contrasts with the isotropic nature of the thermally induced contribution. Please note that we discuss this polarization-dependence thoroughly in the answer to question 3.3. by Reviewer 2, amongst others by providing additional data showing flow generation attempts with linear, elliptical and circular polarization under identical illumination conditions. There we find that under circularly polarized illumination no continuous flow develops, despite the polarization not strongly affecting overall photo-absorption and isomerization (see answer to question 3.1 by Reviewer 2) and the intensity pattern being the same.

Furthermore, for illustration purposes, we would like to show how a continuous photo-thermal flow driven by particle absorption could look instead. We can easily create such a condition in the lab by using non-deformable polystyrene particles, as shown in the figure below. Note that in this case the particles are less easily adsorbed to the interface (for the surfactant-based strategy used in this article see answer to question 2 by reviewer 2) and thus there is a larger ratio of particles in the bulk vs. particles at the interface. In addition, we use extremely high intensity ($>1000 \text{ W/cm}^2$) to obtain a sufficient photo-thermal effect. Upon this highly intense illumination,

particles from the bulk are seen to emerge from the bulk and adsorb to the interface. A slow, constant radial velocity profile (velocities around $1.7 \mu\text{m/s}$), naturally independent from polarization, develops. We note three differences between this illustrative example and the flow phenomenon shown in the manuscript:

- 1) In the example, particles are freely diffusing both during and in the absence of illumination. In our case, individual particle diffusion is easily seen as absent when the illumination is off (see e.g., Supplementary Movie M9). A thermally induced flow would thus have to overcome the particle assembly forces, breaking the capillary assemblies up and at the same time causing a smooth, directional flow, which seems unlikely.
- 2) In the example, the thermally induced flow is radial and independent from the polarization, reflecting the axial symmetry of the Gaussian laser beam (i.e., of the intensity pattern). In our case, axial symmetry is broken with flows following the direction of linear polarization and with circularly polarized light causing no flow at all, all the while the illumination pattern does not change.
- 3) In the example, the flow is continuous because new particles arrive in the illumination zone from the bulk (otherwise the absorbing particles would deplete in the center of the spot). In our case this is not observed, rather the flow gets driven continuously by particles pulled in orthogonally to the polarization/outflow direction. These inflowing particles would have to overcome the very same thermally induced effect. Note that the capillary interaction does not have any preferential direction in the far field as commented at the very end of this reply.

Figure. Isotropic photothermal interface flow using polystyrene particles. a) Initial situation, where few particles are adsorbed to the interface and freely diffusing. b) A while after intense illumination ($>1000 \text{ W/cm}^2$) was turned on, a large number of additional particles can be seen to emerge from the bulk due to photothermal effects and adsorb to the interface at the center of the Gaussian beam laser spot, initiating a weak radial flow. c) The flow is sustained as more particles continuously adsorb to the interface and continue to drive a radial flow, whilst they still freely diffuse. d) When the illumination is shut the particles diffuse at the interface without an overall flow pattern. e) When illumination is resumed, the sustained radial flow emerges again, with the same profile, even if the polarization was rotated from vertical to horizontal. The radial movement is slow and the reference arrows (yellow/gray) stand for $3 \mu\text{m/s}$ velocity. Scale bars (white/black): $10 \mu\text{m}$.

Text modifications: Same as for question 3.3. by reviewer 2, adding supplementary data about flows under linear, elliptical and circular polarization (new Supplementary Figure S12), and highlighting the discussed aspects in the manuscript. We are open to including the discussion of the small transient in-plane isotropic photothermal effect at illumination onset (together with the associated above figure as Supplementary Information) into the manuscript, if judged helpful by the reviewers.

(3) Photoisomerization is another possible driving force for directional movement (e.g., *Advanced Functional Materials*, 2016, 26, 3164). Because a change of polarized light would lead to assembly and disassembly, the authors should clarify the impact of photoisomerization on directional particle assembly.

We are aware that isomerization of azobenzene molecules in principle can lead to phototaxis in bulk liquids or directed motion at liquid interfaces, when isomerization gradients are involved. We believe, however, that the same arguments mentioned in response to question 1 also apply here. Namely, no directed motion can be observed in the sparse particle cases in section 2 which would result from the Gaussian beam illumination gradient (weak over the length scale of the particles). More generally, linear and circular polarization would lead to similar *cis* isomer populations in principle (see answer to question 3.1 by reviewer 2), hence contradicting the very different behavior under the differently polarized illuminations with same intensity and especially the absence of flow for circularly polarized illumination under equal conditions.

Text modifications: To clarify this question, along with the previous ones, an additional paragraph was added to section 3, where the reasons permitting to exclude such effects are elaborated on.

(4) Considering the capillary effect, the mechanism of assembly and disassembly is still not very clear. The particle shape is very important in capillary assembly, but lots of research shows the spheres can also achieve capillary assembly. Why does the shape deformation here lead to a rapid disassembly? Is there an impact of interfacial interaction on stabilizing or destabilizing particle aggregates? Does changing the light from elliptical polarization to linear one or reverse operation have some impact on the assembly or disassembly? If the explanation is the variation of out-of-plane curvature radius as the authors claimed, the particle dimension should play an important role in the capillary-driven assembly and disassembly. The authors should provide information on this impact.

This is a set of highly interesting questions about the capillary processes involved, and we thank the reviewer for raising them.

Reason for rapid disassembly

Regarding the question of why nearly in-plane isotropic, flattened disk-like particles tend to disassemble, we believe this issue is closely related to the question of why pristine spherical particles remain dispersed, despite the presence of capillary interactions between them. This point is discussed in detail in our response to Reviewer 2, Question 2.

In brief, we consider the steric stabilization provided by PVP to be a key factor: it prevents aggregation of spherical particles at the interface and also facilitates disassembly upon particle deformation, by maintaining effective repulsion even in the assembled state. The stabilizing effect appears sufficient to counteract attractive capillary interactions. Although the exact physical mechanism underlying this unexpectedly strong interfacial stabilization by steric layers is not yet fully understood, we are actively investigating this aspect using optical tweezers and hope to clarify it in future work.

Polarization sequences for assembly-disassembly

Regarding possible polarization sequences and corresponding assembly-disassembly processes, the observations listed below are worth mentioning. During their discussion it should be kept in mind that different polarizations lead to similar photo-thermal effects and *cis* isomer populations (see answer to question 3.1 by reviewer 2). All polarizations also lead to continuous flattening of the particles along the z-axis (see answer to question 3.2 by reviewer 2). The main difference is if the in-plane particle expansion happens along one axis (linear polarization), mostly along one axis (elliptical polarization) or along both in plane axes (circular polarization). The observations are:

- 1) First, when directly applying circularly polarized illumination to pristine spherical particles no assembly is produced (see new Supplementary Fig. S7 below), regardless of photo-thermal effects and isomerization effects (which still occur as side-effects of the isomer cycling producing the photo-deformation).
- 2) We point out that to see this disk-like deformation without assembly, the circular polarization had to be very well adjusted. Before we optimized the setup for exact polarization control by placing the waveplates directly under the objective leading to the sample, we always observed assembly even for slight elliptical components of the polarization and of the resulting photo-deformation. In these cases, and without perfectly circularly polarized illumination, we were not able to disassemble the particles by any type of prolonged illumination either.
- 3) In fact, the only sequence that we found that can produce assembly and subsequent disassembly is short elliptical or linear polarization (anisotropically deforming the particles but not too much), followed by prolonged circular polarization (flattening the particles much further and rendering them more isotropic in the plane again). The reason that the first illumination needs to be short is that in a shape-mediated capillary force framework, circular polarization will weaken the capillary forces both by flattening the particles (reducing the attainable height of the undulating contact line) and by reducing the in-plane anisotropy of the particles (reducing the geometrical reason for an undulating contact line). In terms of curvature radii, the first would correspond to strongly reducing the out-of-plane curvature radius while the second one corresponds to reducing the difference between local in-plane curvature radii around the particle. However, the total photo-deformation of the particles is finite (as apparent from the graph in answer to question 3.2 by Reviewer 2) and azopolymer deformations are plastic processes where subsequent deformations superimpose onto each other. Hence the first anisotropic deformation should not be too pronounced for circularly polarized illumination to be able transform the shape into a heavily flattened one with low in-plane aspect ratio. Note that already the disassembly of particles initially deformed into more anisotropic shapes by linear polarization (see e.g., Supplementary Movies M4,M5) is observed to take more time than the disassembly of less anisotropic initial shapes produced by elliptical polarization (see e.g., Supplementary Movies M2,M3). We would in principle have liked to assess these properties quantitatively but see ourselves a bit limited amongst others by not having access to monodisperse size particle sets. Hence, we chose to focus on displaying the conditions that reliably work, in order to elucidate this capillary assembly mechanism to the reader, which is crucial for the further understanding of the manuscript. However, we think that adding the negative assembly result under circularly polarized light to the Supplementary Information is a good idea to further underline the shape influence onto the assembly process.

New Supplementary Figure S7. No assembly with circular polarization. a) Irradiation of pristine particles with circular polarization shown at different timepoints of illumination. The full movie is included as Supplementary Movie M6. Despite a similar density condition to Figure 2c/Supplementary Movie M3 (elliptical polarization) or Supplementary Fig. S6b/Supplementary Movie M5 (linear polarization), no assembly is induced for circular polarization. The particles diffuse freely as their shapes are transformed into flattened ellipsoids. Intensity $I = 219 \text{ W/cm}^2$, Scale bar (white): $10 \mu\text{m}$.

Particle size influence on capillary interactions

The reviewer is also very right about hypothesizing a particle size dependence of the disassembly process. In fact, all capillary forces induced by smooth shapes scale up with the particle size (up to sizes where gravity starts to play a role), since the corresponding (geometrically defined) surface deformations are also scaled up (Botto, L., P. et al., *Soft Matter* 8, 9957–9971 (2012)). Therefore, in our polydisperse particle set, we observe particle sizes playing a role in both capillary assembly and disassembly as shown in the new Supplementary Figure S9 below. First, when occasionally observing the assembly of multi-particle aggregates with particles of strongly varying sizes, one can see the following effect: smaller particles get kicked aside to make the more strongly attractive larger particles assemble next to each other (a).

Therefore, smaller particles tend to already be located on the periphery of large aggregates after assembly as visible in (b), leftmost frame. Upon circularly polarized illumination (towards the right), we then consistently observe that smaller particles will leave the aggregate faster, since they more easily reach a lowering of capillary forces sufficient to disassemble. Note that the latter effect can also already be appreciated for the disassembly process in Figure 2c and in Supplementary Figure S8. Given the polydispersity of the sample, quantifying this information statistically is tricky, as we cannot simply compare the behavior of sets of particles with defined sizes and always have interactions between multiple particles with different sizes. An attempt was made in (c), studying the parallel disassembly of a set of 13 multi-particle aggregates under circularly polarized illumination. We then plot the time (manually detected) at which a particle dissociates from an aggregate vs the size (automatically detected for all particles at a common time point after flattening), giving rise to a clear trend of later disassembly for larger particles. In summary, although capillary forces of course are not the only ones scaling with particle size, this is a further puzzle piece supporting the capillary assembly picture.

Supplementary Figure S9. Particle size influence on capillary assembly-disassembly. a) Example of fast dynamics in a multi-particle assembly process under illumination with elliptically polarized light. The two largest particles in the field of view cause the largest interface deformations, giving rise to the strongest capillary interaction. Initially, the upper large particle leaves its partner to approach the lower large particle (top row). Upon contact it rebounds, before approaching once again, this time with its initial partner attached (bottom row). After rotating into parallel orientations, the two large particles squeeze out a small particle between them and assemble side-by-side, with the small particles located in the assembly periphery. The full sequence is shown in Supplementary Movie M8 (first part). Scale bar (white): 5 μm . Intensity: 219 W/cm². b) Example of the stepwise disassembly using circularly polarized light of a large multi-particle aggregate, which was previously assembled by elliptically polarized light. Due to assembly processes as shown in (a), the smallest particles are typically already located on the aggregate's periphery (left). During the illumination, these small particles dissociate from the aggregate rapidly, whereas larger particles first rotate into a tip-to-tip configuration, before also disassembling (similar to what can be seen on Supplementary Fig. S8). Orange circles highlight particles that just left the aggregate in each frame. Scale bar (white): 5 μm . Intensity: 219 W/cm². c) Overview images showing the fully assembled and fully disassembled states, before and after illumination with circular polarization for 30 s of several assemblies, including the one in b (magenta outline) and the full sequence is shown in Supplementary Movie M8 (second part). On the right, the particle area is evaluated for all particles at the same timepoint (4 s after illumination onset) and plotted against the manually evaluated disassembly time, with the data shown in b colored in magenta. The data shows the values for 89 particles in total stemming from 13 different assemblies and displays a clear trend of increasing particle disassembly time vs. particle area (linear slope 2.9 s/ μm^2 , R² = 0.4). Spread in the data obtained can be attributed to the polydisperse particle interactions, where not only a particle's size, but also the size of the particle it is dissociating from, play a role. The frame used for particle extracting the particles' areas, the employed imageJ procedure, the segmented frame and the extracted values are included in the shared data folder.

Text modifications: We include the counter example of non-assembly of particles deformed directly into disks by circular polarization as a new Supplementary Figure S7 referenced in section 2, to provide a further indication about the particle shape's role in (dis-)assembly processes. We also include the new Supplementary Figure S9 showing the weaker capillary forces for smaller particles (getting kicked out during assembly and leaving early during disassembly) in the same section. In addition, we undertake minor text modifications.

(5) The effective length (or interparticle distance) of the capillary effect in driving directional particle assembly should be clarified.

The effective length of the capillary interaction is usually indicated as up to a few 10's of times the particle diameter (see e.g., Trevenen, S. et al., *ACS Nano* 17, 11892–11904 (2023) and the other citations provided below). Apart from the particle size the interaction length of course also depends on the geometry of the anisotropic particle (aspect ratio, flattening, roughness etc.). Thus, giving a precise answer in our case of polydisperse particles being continuously deformed is challenging. What we can do to give the reviewer/reader an idea is considering the distance over which particle assemblies form (at particle density below the flowing threshold). Applying such a strategy to the particle assembly by short elliptical polarization utilized at the end of the previous question and tracking the initial distances between particles ending up in the same aggregate, we end up with a rough empirical estimate of the maximal interaction distance for this illumination condition (see new Supplementary Figure S5 below). We point out that the maximal initial interparticle distance that would lead to pairwise assembly may be underestimated at this density, since the particles tend to directly assemble with their nearest neighbor. Conversely, the maximal pairwise assembly distance is likely overestimated by multi-particle assembly considerations, since particles in the center of a forming assembly can relay capillary forces to the peripheric particles. In addition, we acknowledge that numerous studies about such interaction distances exist with prefabricated well-defined particle sizes and aspect ratios, for example measuring the interaction force vs distance for particles controlled with optical traps (Lim, J. H. et al., *Langmuir* 34, 384–394 (2018)) or by thoroughly analyzing particle approach trajectories for very dilute dispersions (Loudet, J. C. et al., *Phys. Rev. Lett.* 94, 018301 (2005)). Therefore, we also refer the reader to these studies for more precise information about this aspect of the capillary interactions.

New Supplementary Figure S5. Effective interparticle distances leading to assembly. Example quantifying all initial inter-particle distances leading to mutual assembly after a short (7s), elliptically polarized illumination step with intensity $I = 219 \text{ W/cm}^2$. a) Frame before illumination, with freely diffusing particles labeled with a different color for each of the 19 final, separate particle assemblies shown in (b). c) Plots of the initial distances between particles undergoing pair-wise assembly at an early stage, as well as initial distances between all particles ending up in the same final assembly. Here, pair-wise assemblies are observed up to separation distances of 20 μm and particles initially up to 50 μm apart are found to end up in the same assembly. While these distances are consistent with the orders of magnitude for effective interaction distances reported in literature, note that particle sizes and geometries will influence the interaction length. Hence, e.g., the illumination time and polarization used may influence the final measured result. The same amounts for the time interval up to the

definition of “final assemblies” since assembly events (for example of smaller sub-assemblies) can be observed also after the illumination is terminated. Finally, the particle density can also play a role in defining the initial distances leading to common assembly. In dense particle settings, such as employed for example in Figure 3, the particles can supposedly more easily relay the capillary forces between each other, and the final assembly appears to span large parts of the illumination spot (80 μm) as indicated by its strongly reduced diffusion visible when the laser is off. The full movie used for the empirical estimates shown here, as well as the segmented initial particle position frame and labeled positional values are included in the shared data folder.

Text modifications: The empirical example of interparticle distances shown above is included as new Supplementary Fig. S5 and briefly mentioned in section 2. The above references to the effective interaction lengths in the literature were included.

Further Comment: We decided to not use the term directional particle assembly further, because we realize it may lead to misunderstandings (one occurrence removed from section 2 and changed into ordered capillary assembly). In fact, what is directional is the photo-deformation. Once particles are deformed, they assemble into ordered (e.g., side-by-side structures), but they may rotate in doing so (see Supplementary Fig. S4, further descriptions e.g., in Trevenen, S. et al., *ACS Nano* 17, 11892–11904 (2023) and nice visualization in Loudet, J. C. et al., *Europhys. Lett.* 85, 28003 (2009)). It is important to mention that the surface deformation which the particles minimize by assembling can be described as quadrupolar in the far field (Loudet, J. C. et al., *Phys. Rev. Lett.* 97, 018304 (2006)). Thus, y -axis deformed particles will feel the same attractive capillary potential when placed with a sufficient distance on a y -oriented line vs. at the same distance on a x -oriented line (no global directionality). During the flow phenomenon, we observe that the directionality is given by the direction of instant particle photo-deformation, rather than the initial orientation of the deformed particles, which becomes apparent from the instant flow direction switching when the polarization is rotated during flowing (see Supplementary Movie M11).

Reviewer #2 (Remarks to the Author)

The revised manuscript has been enhanced and is now much clearer than the original version. The authors have meticulously addressed my comments one by one. This version effectively clarifies that the continuous flow was not a direct result of traditional nano-locomotion due to photoisomerization, but rather from the ongoing self-assembly of ellipsoids when exposed to a linearly polarized light. These self-assembling movements then consistently expelled water in the same direction as the long axis of the self-assembled clusters. The concept of a novel driving force for particle flows proposed in this study may engage readers.

However, concerns have emerged regarding the adaptability of this strategy due to several limitations. Furthermore, the discussion in the concluding paragraph lacks persuasiveness in its assertions about the strategy's impact.

REPLY: We thank the reviewer for the positive assessment of the improved clarity of the manuscript. Regarding the adaptability of the findings, we added a broader perspective in response to the specific points raised below.

First, the functionality of the particle is notably constrained. The range of applicable materials is predominantly confined to DR1 polymers, characterized by a substantial overlap of the trans-cis bands. In the absence of this overlap, conventional nano-locomotion mechanisms are likely to be activated. This limitation in material applicability restricts the functional diversity of the particles. Additionally, the functional modification of particle surfaces is impeded because the balance between hydrophilicity and hydrophobicity on the particle surfaces is vital for achieving a dense distribution on the water surface while simultaneously preventing the initial formation of aggregates. The functional limitation constrains the potential application of this strategy.

REPLY: We realize that the variety of effects shown in this work, which can be controlled by comparatively simple optics, may have sparked the reviewer's curiosity about adaptability to other materials. Traditionally, the azopolymers used for polarization-dependent photo-deformation indeed often contain DR1, although similar pseudo-stilbene-type azobenzene dyes such as DR19, Disperse Orange 3 and others have also been used (see e.g., Yin, X. et al., *Macromolecular Chemistry and Physics* **2018** 219, 1800113, Jo, W. et al., *ACS Appl. Mater. Interfaces* **2020** 12, 5058–5064 and Pirani, F. et al., *Scientific Reports* **2016** 6, 31702). In the case the reviewer's interest is directed towards more conventional azobenzene-type dyes (without strong overlap of the bands), the simultaneous use of two wavelengths may be considered to achieve enhanced cycling between the isomers (a strategy utilized e.g., in Liu, D. & Broer, D. J., *Nat Commun* **2015** 6, 8334). In this case, a trans-to-cis wavelength (e.g., UV) and a cis-to-trans wavelength (e.g., blue light) are applied simultaneously to effectively induce isomer cycling also for such dyes. We believe that the trans-to-cis irradiation should then be polarized, if re-orientational effects are to be studied.

In the context of this work, since all the effects ultimately rely on the anisotropic photo-deformation of particles, *employing anisotropic photo-deformable particles is the minimum criterion for adaptations*. We point out that numerous other approaches to obtaining anisotropic photo-deformable particles exist. They do not necessarily rely on molecular re-orientation through isomer cycling and, in some cases, do not even require the presence of azobenzene molecules. Examples include liquid crystal elastomer (LCE) particles (see e.g., B. Braun, L., Hessberger, T. & Zentel, R., *Journal of Materials Chemistry C* **2016** 4, 8670–8678), as mentioned in the Discussion section. LCEs are directionally deforming materials, whose photo-deformation in some cases can also be polarization-dependent (Yu, Y., Nakano, M. & Ikeda, T., *Nature* **2003** 425, 145–145), particularly if dichroic dyes are included (Li, Y., Liu, Y. & Luo, D., *Adv. Optical Mater.* **2020** 9, 2001861). Other polymeric materials, for example hydrogels doped with anisotropic platelets, can also be photo-deformed anisotropically (Zhu, Q. L. et al., *Advanced Materials* **2024** 36, 2314152) and may therefore be used for similar concepts in future studies.

Finally, the particles need to remain colloidally stable and efficiently adsorb to the liquid interface. In this work, we employed steric stabilization, as it simultaneously ensures colloidal stability and promotes interfacial

adsorption (Rey, M. et al., *Nat Commun* **2022** *13*, 2840). The energy gain upon colloidal particle adsorption to a liquid interface is on the order of millions of kBT; therefore, at this length scale, the balance between hydrophilicity and hydrophobicity can almost be ignored, and particles generally adsorb (Pieranski, Pawel. *Physical review letters* **1980** *45*, 569). Nevertheless, kinetic barriers can impede monolayer formation, and various additives have been reported to facilitate particle adsorption (Vialeto, J. & Anyfantakis, M., *Langmuir* **2021** *37*(31), 9302–9335). Moreover, several other strategies exist to maintain colloidal stability once particles are adsorbed at the interface (Vogel, N. et al., *Chemical reviews*, **2015**, *115*(13), 6265–6311). A widely used approach involves charge-stabilized colloids. Depending on the balance between attractive van der Waals and capillary interactions and repulsive dipole–dipole and electrostatic interactions, stable non-close-packed monolayers can be obtained (Menath, J. et al., *Advanced Materials*, **2023**, *35*, 2206593.) In summary, also with respect to functional modifications, we see no fundamental reason why the reported results could not be extended to other colloidal particle systems.

Second, the lack of correlation with light intensity, which was stated in the rebuttal, hinders the precise quantitative control of directional flows. Although the authors suggest the potential application of various light properties, controlling directional propulsion by intensity gradients remains unlikely.

REPLY: We are not entirely certain if we understand the reviewer’s concern correctly. What we stated in the rebuttal letter and pointed out additionally in the manuscript, is that the intensity pattern does not define the flow *alone*.

Specifically, different flow scenarios can be created by changing polarization while keeping the intensity pattern constant. For example, using the same Gaussian laser profile, one can obtain no flow (circular polarization), weak directional flow (elliptical polarization), and strong directional flow (linear polarization), where the directionally axis can be rotated for the latter two.

If the polarization is kept constant instead, and the intensity pattern is altered, this provides an additional means to control and alter the flow. For example, a 45-degree linear polarization of a Gaussian beam intensity profile leads to the symmetric 45-degree oriented outflow in Supplementary Movie M11. In contrast, when a stripe-shaped intensity pattern is used with the same polarization, a pure shear flow is created (Supplementary Movie M15), demonstrating that combinations of intensity and polarization patterns can be deterministically exploited to control particle flows. In addition, higher nominal intensities – with otherwise identical patterns – create faster flows, since the particles’ deformation is accelerated (see example in the Figure below).

Hence, in contrast to flows driven by intensity patterns only, where each point on the interface can, in theory, be attributed a given intensity, here each point on the interface can be attributed both *an intensity and a polarization*, together defining the strength and the directionality of the locally created flow contribution. This allows the flow patterns to be altered through modification of *either* intensity patterns *or* the polarization. Considering this, a mere correlation of light intensity with the “quantitative control of directional flow”, would not fully capture the complexity of the effect observed. We have added a clarifying sentence towards the end of result section 4 in the manuscript to highlight this aspect more explicitly and further elaborated on it in the revised Discussion section.

Figure. Role of illumination intensity. a) Overlay of the symmetric flow pattern obtained using a Gaussian spot illumination with linearly y-oriented polarization as shown in Figure 4c (light blue arrows) and flow pattern elicited with the same intensity pattern and polarization, but at reduced nominal intensity (purple arrows). Scale bar (black): 20 μm . Velocity reference arrow (grey): 20 $\mu\text{m/s}$. b) Same as in (a), but for the shear flow pattern obtained using a stripe illumination pattern and diagonal linear polarization as shown in Figure 4g (light blue arrows) and a corresponding reduced intensity version (purple arrows). Scale bar (black): 20 μm . Velocity reference arrow (grey): 30 $\mu\text{m/s}$. Note that while the velocities (i.e., the arrow length) decrease for lower nominal intensity illumination, the overall flow pattern (defined by the relative intensity distribution and polarization) is maintained. Raw movies of the reduced intensity flows are included in the shared data folder ('//Flow_Raw_Movies/016_Xtra...' and '//Flow_Raw_Movies/017_Xtra...').

Moreover, the nonspherical shapes of the particles are insufficient for promoting diverse particle flows, as the generation of capillary forces requires one-dimensional deformation of the particles, irrespective of their initial shape. Nonetheless, the authors have proposed the potential application of buckling particles. Additionally, the use of Janus particles would interfere with the one-dimensional stretching of the particles.

REPLY: This comment appears to reflect a slight misunderstanding, and we apologize for any lack of clarity in our original explanation. The aim of that paragraph was not to suggest that our system could be combined with these other approaches (buckling particles, Janus particles), but rather to illustrate that any directional flow needs a symmetry breaking element (making the flow directional). In different systems, different light properties (such as intensity gradients, propagation direction, or polarization states) can serve as this symmetry-breaking factor. We have rephrased this second-last Discussion paragraph accordingly.

The findings of this study must demonstrate substantial potential for future advancements that surpass the objectives delineated by the authors in the Discussion section.

REPLY: We appreciate the reviewer's encouragement to emphasize the broader potential of this work. While we agree that it would be highly interesting to demonstrate "substantial potential for future advancements", we believe that this cannot be convincingly demonstrated *a priori*. Instead, we have aimed to outline promising development directions based on the newly observed effect and proposed application domains in which these findings may prove beneficial. It should be noted that further advancements may rely both on the capillary assembly mechanism itself and on the ensuing flow phenomenon (addressed in separate Discussion paragraphs). Potential routes toward improved and novel behavior may involve different functional materials (such as e.g., LCEs for reversible photo-deformation; see the second Discussion paragraph) and/or refined experimental scenarios (such as e.g., polarization patterns or passive objects interacting with the flows; see final Discussion paragraph).

The following are additional minor comments:

1. Although stretched particles predominantly self-assemble in a side-by-side manner, the formation of end-to-end configurations is also essential for the development of elongated assemblies. The end-to-end configurations are clearly illustrated in Figure 3c, panels iii and iv, along with the side-by-side configurations. The mechanism underlying the formation of the end-to-end configurations must be elucidated.

REPLY: We thank the reviewer for drawing attention to this detail. For ellipsoidal particles at liquid interfaces, both side-by-side and end-to-end (tip-to-tip) configurations are energetically favorable, since each reduces the overall interfacial deformation by superimposing matching surface elevations and depressions of the adjacent particles. In contrast, configurations such as side-to-tip are energetically disfavored. In the case of core-shell particles, such as the colloidal particles with a PVP shell investigated here, the energy gains associated with the two assembly modes are comparable and depend on the particle aspect ratio and the core-to-shell size ratio (Eatson, J. et al., *Journal of Colloid and Interface Science* **2025** 683, 435–446).

For particles with a high aspect ratio, the side-by-side configuration yields a greater energy reduction than the tip-to-tip configuration and is therefore typically the preferred configuration when the interfacial particle density is low (as in the experiments of the first two result sections). At higher interfacial particle densities, additional energy can be minimized by forming tip-to-tip contacts between particles already arranged side-by-side, a process that has been described both experimentally and theoretically (Luo, A. M. et al., *Journal of Colloid and Interface Science* **2019** 534, 205–214). Qualitatively, we observe the same trend of more tip-to-tip assembly in our experiments at higher interfacial particle density (see Figure 3).

In addition, we note that the capillary interactions between anisotropic particles are strongly attractive, and thus the assembly process can easily become kinetically trapped in local minima when many particles are involved. Together with particle size polydispersity in our system, this likely explains the large structural variability observed at high particle density and the frequent occurrence of tip-to-tip bonds in the elongating assemblies. We have added a sentence mentioning this observation in result section 3.

2. The head-tail configuration of the DR1 main chain structure depicted in Figure 1 is inverted. The sequence -C(CH₃)-CH₂- needs to be altered to -CH₂-C(CH₃)-.

REPLY: We thank the reviewer for the careful reading and have corrected the graphical typo.

3. The section headings should be revised. Most of the discussion is detailed in the Results section, while the Discussion section primarily summarizes this manuscript.

REPLY: The reviewer correctly notes that the technical discussion of the different effects is presented directly in the result sections. This structure was chosen to avoid confusion between the different novel effects shown in the different sections and not to leave the reader with too many open questions along the way.

In the final Discussion section, we try to contextualize the results more broadly, including a concise summary and several outlook perspectives. According to our understanding of the journal's formatting guidelines, this final section should indeed be labeled as "Discussion". In either case, we would ultimately prefer to consult the editor regarding this stylistic decision, if possible.

4. Considering the title and the focus outlined in the Introduction, it is inappropriate to conclude this manuscript by emphasizing the relevance of tiled incidence illumination to conventional nano-locomotion systems in the closing paragraph.

REPLY: As outlined above, the aim of this section is not to directly compare different strategies in terms of performance (which we believe depends on application scenarios and on potentially refined systems), but rather to show that any directional flow system requires an effective strategy for symmetry breaking. The referenced articles are some of the most elegant and impressive examples of different optical symmetry-breaking

mechanisms that rely on various properties of light (for example, the incidence direction). We then point out that polarization has not been used frequently as a symmetry breaking mechanism, yet strategies based on this approach—or similar ones—may hold significant potential and could be adapted to many other systems. Strategies relying on polarization and polarization-directed deformation may differ substantially from the specific one presented here and might not even rely on capillary forces to collectively propagate deformation. This realization, as well as the possibility to further combine intensity and polarization patterns to obtain more complex flow fields, has been added to the Discussion section (last two paragraphs), to broaden the adaptability perspective before mentioning possible applications of the flow concept.

Reviewer #3 (Remarks to the Author)

The reviewer thanks the careful response and additional experiments for the revision. The response answers some questions. However, some content still sounds confusing.

REPLY: We thank the reviewer for the positive comments and the constructive feedback. In the following, we will address the remaining open questions in detail, particularly with respect to thermal effects.

1. Photoisomerization has shown a crucial effect on particle deformation. And photothermal conversion would influence the surrounding flow. The authors did not do the characterization. The UV-vis spectra and photothermal curves should be given to reflect photo-responsive behaviors.

REPLY: Photoisomerization: The work is performed using a commercial compound and the spectroscopic properties in question are well known. *Trans* and *cis* spectra of DR1 can be found in *J. Phys. Chem. A* **2006**, *110*, 11926-11937 (Figs.2 and 3). More specifically, when looking at Fig.3 in the above referenced article, we observe that the spectral absorbance change due to trans-cis isomerization agrees well with the relative transmission changes during and after exposure to laser light that we measured and attached to the previous rebuttal letter, i.e. an increase of transmission at 570 nm and a decrease of transmission at 600 nm associated to trans->cis excitation. The time dynamics also align well with what has been previously reported for the same DR1 dye, and even for similar acrylic polymer systems incorporating this dye (e.g., in Barrett, C. et al., *Macromolecules* **1994** *27*, 4781–4786 and/or in Barrett, C. et al., *Chem. Mater.* **1995** *7*, 899–903). Re-orientational effects and directional photo-deformation arising from continuous trans-cis-trans cycling in these dyes are well summarized in Natansohn, A. & Rochon, P., *Chem. Rev.* **2002** *102*, 4139–4176, and more recently in Saphiannikova, M. et al., *Soft Matter* **2024** *20*, 2688–2710.

We think that repeating these well-known characterizations that are reported in literature in the past 30 years in further detail is out of scope of this work. As for the feasibility of such experimental measurements, we note that this would require modifying an existing photo-spectrometer by adding a polarization-controlled laser source — an inconvenient adjustment for instruments and equipment in shared facilities. Again, we do not see the need for this. We already provided measurements of the transmission change due to the photo-isomerization at two target wavelengths. The results match the expected behavior described in referenced works.

Photo-thermal behavior: To assess the potential impact of laser-induced heating of the polymer in a controlled and realistic setting, we employed the setup shown in the Figure below (a), wherein a doubled-frequency Nd:YAG laser beam is injected into a single-mode fiber whose terminal facet is directly placed in close contact with the thin (120 μm) coverslip protecting the samples to be measured. Temperature distributions are then directly collected using a thermal camera on the opposite side of the sample. Here, the samples consist of azopolymer particles from the manuscript (b) and continuous azopolymer films (c,d) which were spun and/or dried on the protective coverslip. In either case, the coated side faces a $\sim 100 \mu\text{m}$ gap (with parafilm as spacer) between the sample and another coverslip, glued such that the gap can be filled with water by capillarity and to wet the azopolymer sample. In this way, a stable situation for thermal measurements can be obtained, with a realistic cooling component of the aqueous phase and easy integration of the visible light illumination with the infrared thermal imaging (different optics). For all measurements, the illumination power was 60 mW, which results in an equivalent power density of no less than 339 W/cm^2 , assuming a mode field diameter of at least 150 μm as a reasonable upper-limit estimate of the illuminating spot size, after propagation from the fiber end to the sample surface (see fiber specs here https://www.thorlabs.com/newgrouppage9.cfm?objectgroup_id=1362). In these conditions, we operate at higher power densities than needed to induce flows observations with floating particles at water/air interface.

We first evaluated the sample shown in (b), which consists of the same azopolymer particles used throughout the manuscript. These particles were drop-cast onto the substrate. During drying, the azopolymer particles adsorb at the liquid–air interface of the droplet (Rey, M. et al., *Nat. Commun.* **2022**, *13*, 2840), where the packing density closely matches that observed at the liquid interface in the main experiments (Figs. 3 and 4). Once the

interface is fully covered with a particle layer, the remaining particles undergo the characteristic coffee-ring deposition (Rey, M. et al., Nat. Commun. **2022**, *13*, 2840). As a result, the dried droplet shows a dense, 3D-stacked particle ring along the perimeter and a nearly continuous monolayer of particles in the center.

In the central region, this monolayer thus reproduces the interfacial particle arrangement from the main experiments (worst case). We then used this area to assess potential laser-induced heating. However, we did not observe any detectable temperature changes between the laser on and laser off states, except when moving over the particle multilayer “coffee ring” at the boarder of the sample (see real photographs as insets). Even there, the detected temperature difference between the two stationary situations remained below 1° C.

To obtain a more well-defined, though more extreme, worst-case scenario, we then spun pure azopolymer from anisole solutions onto glass slides (as shown in c,d). In this case, the sample consisted of a continuous film of azopolymer, which, in (d), was measured to be approximately 3.5 μm thick using a profilometer, i.e., thicker than the diameter of the largest particles in this work. Here, we were able to observe a larger maximal temperature change, but it was still below 3° C. We consider this estimate of the temperature change to present a very generous upper limit, given the thickness of the film (larger than the particles used), the intensity employed (in the upper range of what is used in this work) and the continuous nature of the azopolymer film (as opposed to the non-continuously dispersed particles in the water-air interface experiments). Note that what is lacking compared to the flow experiment situation is a possible heating contribution of bulk-dispersed particles, which, however, should be small in the actual experiments due to the efficient interface adsorption (see also answer to question 4 below).

Figure. Thermal measurements. a) Photograph of the setup used to estimate the photothermal behavior of the samples during the experiments. A fiber couples in the green visible light (60 mW) on one side and a thermal camera collects thermal images on the opposite side of test samples. Samples consist of an azopolymer coating deposited on one of two coverslips forming a 100 μm wide cavity filled with water by capillarity during measurements. b-d) Thermal images as acquired by the thermal camera with the mean stationary temperature in the laser-exposed sample zone indicated during stationary laser OFF (left) and laser ON (right). The bottom left inset on each thermal image shows photographs of the samples and how they are positioned on the thermal

images. Samples to be tested consist of dried azopolymer particles in (b) and of continuous azopolymer films in (c) and (d), with a thickness of $\sim 3.5 \mu\text{m}$ measured by profilometer for the thicker sample shown in (d).

To summarize, from the thermal camera measurements, no measurable temperature rise was found for single interfacial particle layers. Although a small degree of photothermal heating is expected in principle, its effect must be very limited. We occasionally observe minute, outward-directed flows immediately upon laser illumination (see previous rebuttal letter, answer to question 2 by reviewer 3) — possibly reflecting this minor local heating — but these flows are negligible under the conditions relevant to the reported particle dynamics. We therefore conclude that photothermal heating is present but too weak to play a significant role in the observed directional flow phenomena.

In addition, the following two key experimental observations are found to be inconsistent with purely photothermal effects:

- Strong dependence on the polarization of the incident beam. Even if the intensity distribution of the illumination is axis-symmetric (e.g. Gaussian profile), the corresponding deformation and particle flow is not axis-symmetric and depend on the polarization state (e.g. for circular polarization, particles flatten to disk shapes and no flow is observed).
- Characteristic time scales for starting the particle flows. As shown in Figure 3d and discussed previously (e.g. in the answer to question 3.7 by reviewer 2 in the previous rebuttal letter), the flows can rise slowly over timescales of ~ 20 seconds when illumination is turned on for the first time (since the particles first have to deform and assemble), but resume instantly at full speed upon subsequent illumination onsets (when the particles are already deformed and assembled). For thermal effects, the rise-time should always be similar.

Taken together, the pronounced polarization dependence, characteristic time scales, and absence of measurable heating unambiguously demonstrate that the observed flows are driven by anisotropic photo-deformation and not by photothermal effects.

2. If the reviewer understands correctly, the authors insist that there is a similar photothermal effect and photoisomerization in response to different polarization lights, so they would not give different influences on the flow. The change of polarization light would induce particle deformation into different shapes. The photothermal conversion within different-shaped particles would significantly influence the surrounding temperature distribution and flow direction (e.g., *Lab Chip*, 2012, 12, 3707). The authors should clarify that on particle locomotion at the interface.

REPLY: The reviewer correctly summarizes that similar amounts of photo-isomerization and photo-thermal effects would naturally be assumed for illumination with different polarizations of the DR1 dye. During the peer review process, this equivalence has been explicitly confirmed with respect to photo-isomerization effects (see the answer to reviewer 2's question 3.1 in the previous rebuttal).

However, we struggle to follow the reviewer's hypothesis that the shape of the particles itself would strongly affect their photo-thermal conversion and the surrounding temperature profile. The referenced article (Furlani P. et al., *Lab Chip*, 2012, 12, 3707) seems to explore the photo-conversion within nanometer-sized metallic nanoparticles *exhibiting plasmonic resonances* under illumination with *ultrashort (nanosecond) laser pulses*. While the enhanced optical field around the tips of a nanorod — shown e.g., in Figure 4 of the article — is indeed caused by the interplay of the polarization of the field with the anisotropic particle shape, these effects require the particles to be both sub-wavelength in size and metallic (e.g. Ag or Au) in order to exhibit such plasmonic resonances. When the photothermal conversion resulting from the plasmonic resonance occurs during a short (3 ns) laser pulse with high intensity ($7\text{mW}/\mu\text{m}^2 = 7 \cdot 10^5 \text{ W}/\text{cm}^2$), the authors simulated preferential bubble nucleation around the nanorod tips during the ultra-fast transient heating.

We do not see any meaningful connection between the situation described in the referenced article and the *continuous, orders of magnitude less intense illumination of micro-sized polymer particles* employed in this work.

3. Because the particle shape would influence the surrounding flow direction, the particle assembly into aggregates may further intensify this effect on the flow, leading to directional flow. The authors should clarify that.

REPLY: As discussed above, the question is not directly relevant to this work for the same reasons outlined under the previous point, namely that no anisotropic photothermal mechanism is expected for the non-plasmonic, micron-sized polymer particles investigated here.

Nevertheless, to address this aspect as completely as possible, we considered the reviewer's hypothetical scenario in which a shape-dependent photothermal effect could influence the flow direction. To exclude this possibility, one may consider an experimental situation where previously linearly deformed particles are illuminated with circular polarized light. In fact, below we provide an example of flow during alternating 15 second illumination steps with circularly and linearly polarized light (included as NEW Supplementary Fig. S14). In this case, we see that:

- 1) First, upon initial circularly polarized illumination, the particles become disk-like and no persistent flow develops, as expected and previously shown.
- 2) Secondly, upon subsequent linearly, x-axis polarized illumination, the particles are stretched along the x-axis, assemble and a persistent x-oriented outflow develops, as expected and previously shown.
- 3) Thereafter, when another circularly polarized illumination step is applied, no flow develops, *despite the particles already being assembled and stretched along the x-axis. This directly contrasts with the hypothetical scenario proposed by the reviewer, in which previously inscribed shapes should influence the polarization-dependent flow behavior.*
- 4) Upon the next linearly x-polarized illumination step, the persistent flow is observed again, as usual.

This experiment provides direct evidence that the directionality of the observed continuous flow cannot arise from heating of elongated particles itself but instead originates from anisotropic photo-deformation.

New Supplementary Figure S14. Alternating circularly and linearly polarized illumination steps. a) Flow fields before (left) and during (towards right) sequential illumination steps with an alternatingly circularly and linearly x-oriented polarized laser. Only during illumination steps with linear polarization does a persistent outflow develop. Scale bars (black): 20 μm . Velocity calibration arrows: 5 $\mu m/s$. Time t indicated in the upper right of the flowfields reports total movie time. b) Frames from the corresponding movie which is included in the shared data under ‘//Flow_Raw_Movies/015_Alt_CircHorizPol_50fps.avi’. The yellow rectangle tracks a prominent particle with easily appreciable deformations, which is shown in magnified insets in (c), during the sequential illumination steps. Scale bars (white): 20 μm in (b), 2 μm in (c). Notably, during illumination step 3, no persistent flow develops despite the particles being strongly pre-deformed along the x-axis after illumination step 2.

In addition, I have also looked over the responses to the comments from Reviewer 1.

4. I agree with Reviewer 1. The current microscopy images cannot accurately localize the particles at the interface. And the authors did not give direct evidence for that in the response. It is better to use another characterization instrument (e.g., confocal microscopy with z-stacking scan) to give the particle distribution in the solution. That would also rule out the possible influence of the thermo-flow induced by the dispersed particle underwater.

REPLY: We are slightly surprised that the reviewer does not consider the previously mentioned observations – notably the absence of out-of-plane translational as well as out-of-plane rotational diffusion of particles during the full duration of all experiments – conclusive to this matter. However, we agree that the suggested z-stack method may provide a more intuitive visualization of the interface adsorption to the reader and additionally permits to qualitatively assess the ratio of adsorbed particles to bulk particles. As shown below (in New Supplementary Figure S2), when performing the suggested experiment, the vast majority of particles is found to lie within the same focal plane, corresponding to the water–air interface. This confirms that the majority of particles are adsorbed at the interface rather than dispersed in the bulk phase.

As a side note, we point out that the confocal images were acquired using the intrinsic fluorescence of the azopolymer particles under excitation at 514 nm, corresponding to their absorbance peak. Because this wavelength also induces photo-deformation and the laser in the confocal setup is inherently linearly polarized, the interface-adsorbed particles gradually deform and assemble after several scans (see images on the left).

New Supplementary Figure S2. Confocal stack of interface-adsorbed particles. a) Single 2D slice from a confocal scan, imaging the particles based on the azopolymer's intrinsic fluorescence, at the z-position of the water-air interface. Top: transmitted light detector. Bottom: fluorescent (confocal) signal. b) Full confocal z-stack through the sample cell illustrating that the majority of particles are confined at the liquid interface. Right: schematic illustration showing the different samples sections corresponding to the z-stack depth, with particles illustrated in green. All scale bars (white): 10 μm . Note that the particles become directionally deformed during imaging as described in the result section "Shape-morphing particles at an air-water interface", here in the y-direction corresponding to the linear polarization orientation of the confocal laser that is used for excitation (514 nm). The directional deformations lead to capillary bonds as described in the result section "Optically controllable capillary interactions and assembly". These effects are particularly visible on the transmitted light image, whilst the presence of smaller particles (see Supplementary Fig. S1) is more easily appreciated on the fluorescent one.

5. The authors discussed the influence of surface coverage on particle assembly and disassembly in the revision. According to Reviewer 1's comments, the flow phenomenon of these dilute systems should be explored. Normally, there should be a thermal Marangoni effect to propel photothermal particles at the interface. In terms of high-intensity light used in this study, it is very surprising that the authors' response indicated there was no flow observed for the diluted systems. The authors should explain this phenomenon. That would be a good control in this study.

REPLY: As listed previously, there are 8 movies (Supplementary Movies M1-M8) showing dilute systems of particles being irradiated in comparable settings to the flow movies (intensity, Gaussian beam width). In addition, we provide 60 movies showing the continuous deformation of dilute, single particles during prolonged illumination with the same Gaussian beam (see shared data '//Particle_Deformation/Movies/', used for Supplementary Fig. S4). In none of these movies is persistent motion comparable to the directional flows detected

(note that there can be slight global shifts upon illumination onset, which are reversed at illumination off). Therefore, we consider the absence of persistent gradient-related photo-thermal particle motion a direct experimental observation.

To understand why no such flow occurs under these conditions, one may consider the relevant length scales. As previously discussed (e.g., in the answer to question 3.3 by reviewer 2, previous rebuttal letter), the relative size of movable objects and of the illumination beam profile matters just as much as the net absorbed intensity itself. In many of the gradient-related motion articles discussed, the beam width is comparable to the object that is displaced, sometimes even combined with a tilted beam (see Lucchetta, D. E. et al., *AIP Advances* **2015** 5, 077147, Norikane, Y. et al., *CrystEngComm* **2016** 18, 7225–7228 or Paven, M. et al., *Advanced Functional Materials* **2016** 26, 3199–3206). Therefore, there can be a 100% difference in photo-absorption from one end of the object to the other. In our case, the Gaussian beam (waist $\sim 70\text{-}80\ \mu\text{m}$) is much wider than even the largest particles ($<3\ \mu\text{m}$), and the incidence is normal. The variation in photo-absorption across particles is therefore only a few percent at most. For deformed particles the relative difference could be slightly higher, however these also absorb much less of the provided intensity due to being very thin.